# Adaptive Physics-informed Neural Networks: A Survey

**Edgar Torres**                                          *edgar.torres@ki.uni-stuttgart.de*
*Institute for Artificial Intelligence*
*University of Stuttgart*

**Jonathan Schiefer**                                     *jonathan.schiefer@de.bosch.com*
*Robert Bosch GmbH, Stuttgart*

**Mathias Niepert**                                       *mathias.niepert@ki.uni-stuttgart.com*
*Institute for Artificial Intelligence*
*University of Stuttgart*

**Reviewed on OpenReview:** *https://openreview.net/forum?id=vz5P1Kbt6t*

## Abstract

Physics-informed neural networks (PINNs) have emerged as a promising approach to solving partial differential equations (PDEs) using neural networks, particularly in data-scarce scenarios, due to their unsupervised training capability. However, limitations related to convergence and the need for re-optimization with each change in PDE parameters hinder their widespread adoption across scientific and engineering applications. This survey reviews existing research that addresses these limitations through transfer learning and meta-learning. The covered methods improve the training efficiency, allowing faster adaptation to new PDEs with fewer data and computational resources. While traditional numerical methods solve systems of differential equations directly, neural networks learn solutions implicitly by adjusting their parameters. One notable advantage of neural networks is their ability to abstract away from specific problem domains, allowing them to retain, discard, or adapt learned representations to efficiently address similar problems. By exploring the application of these techniques to PINNs, this survey identifies promising directions for future research to facilitate the broader adoption of PINNs in a wide range of scientific and engineering applications.

## 1 Introduction

Advances in machine learning have led to important applications in various fields, such as computer vision (enabling technologies like self-driving cars), natural language processing (powering intelligent agents and chatbots), and image generation (facilitating media creation). Motivated by this success, there has been growing interest in developing Machine Learning (ML) solutions to solve problems in science and engineering. Unlike other fields where data is abundant or easily obtained, however, science and engineering often face data scarcity due to the high costs associated with generating data through expensive experiments or simulations. Therefore, to facilitate the development of ML approaches in these disciplines, AI methods that are data-efficient and computationally efficient need to be created. To this end, other domains have tackled similar problems with techniques such as transfer learning, meta-learning, and few-shot learning, indicating significant potential for applying these techniques in the context of science and engineering.

One specific application in science and engineering where these efficient ML models can be particularly beneficial is to determine the approximate solutions of PDEs. PDEs are fundamental for modeling and describing natural phenomena in various scientific and engineering domains. Traditionally, these equations are solved numerically, which can become prohibitively expensive, especially when dealing with nonlinear and high-dimensional problems (Han et al., 2018). This challenge limits their application in areas where

a fast evaluation of a PDE is required. Recognizing this challenge, neural networks have been explored as a potential solution, offering advantages in effectively modeling complex nonlinearities (Raissi et al., 2019; Khoo et al., 2021; Sirignano & Spiliopoulos, 2018; Cuomo et al., 2022), presenting the potential for faster evaluation compared to classical iterative solvers, as well as offering mesh-free solutions not constrained to computational grids (Jiang et al., 2023; Li et al., 2020; Raissi et al., 2019; Cuomo et al., 2022). Moreover, machine learning techniques provide an approach to solving inverse problems, where the goal is to infer unknown parameters or initial/boundary conditions from observed data, a task challenging for numerical methods (Arridge et al., 2019; Cai et al., 2021). In addition, machine learning implementations are simpler than numerical methods, allowing faster development and easy maintenance (Cai et al., 2021).

ML methods for approximating PDE solutions can broadly be categorized into two types: neural surrogates and neural PDE solvers[1][2]. Neural surrogates, including physics-guided neural networks (Faroughi et al., 2023) and neural operators, function by training neural networks to predict data generated from numerical solvers. The most popular among these are neural operators, which approximate nonlinear mappings between infinite-dimensional function spaces using datasets of input-output pairs from solvers or observations. Examples include the Fourier Neural Operator (Li et al., 2020) and DeepONet (Lu et al., 2019). On the other hand, neural PDE solvers directly incorporate physical laws by embedding the governing equations into the learning process. A key example is PINNs (Raissi et al., 2019), which approximate solutions by minimizing the residuals of the governing equations, the initial conditions, and the boundary conditions. Figure 1 illustrates the relationship between data requirements and scientific knowledge between different methods.

Considering the challenges associated with data acquisition in the scientific and engineering domains, this study suggests the use of PINNs to solve such problems. Neural operators typically require large datasets, often derived from costly simulations, and do not explicitly incorporate governing physics equations, which can lead to generalization problems and physical inconsistencies outside the training data distributions (Négiar et al., 2022). In contrast, PINNs integrate governing equations directly into the training process, ensuring that the solutions adhere to the underlying physics while reducing the reliance on pre-existing datasets, making them particularly effective for data-scarce applications (Négiar et al., 2022).

Nevertheless, PINNs have some known limitations. They can struggle with convergence, particularly for complex or high-dimensional physics problems (Négiar et al., 2022), resulting in long training times. The computation of residuals, which requires derivative evaluation by automatic differentiation, becomes computationally expensive for PDEs with higher-order derivatives. Additionally, PINNs' convergence is sensitive to hyperparameters. Furthermore, PINNs are typically trained on a per-PDE

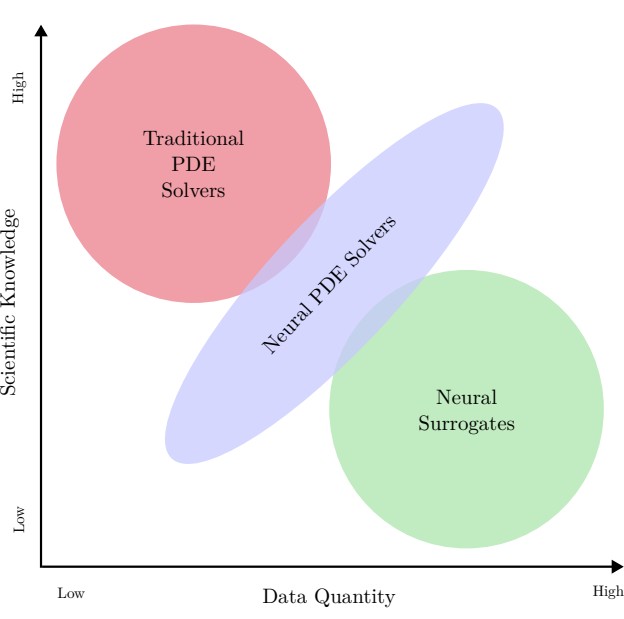

Figure 1: Data and scientific knowledge requirements for different modeling approaches.

instance basis, meaning they can only solve one specific problem at a time and must be re-trained from scratch for each change in parameters. These drawbacks hinder their adoption in diverse tasks that involve fast evaluations of different PDEs.

---

[1]A surrogate model can be thought of as a "regression" to a set of data, where the data is a set of input-output parings obtained by evaluating a black-box model of a complex system Eason & Cremaschi (2014); Caballero & Grossmann (2008). In contrast, a solver is an algorithm or method used to find a solution to a mathematical model.

[2]While some authors use the terms "Neural Surrogates" and "Neural PDE Solvers" interchangeably, this work makes a distinction to highlight the specific requirements for obtaining the solution to a PDE.

To address these limitations, this survey explores the integration of advanced ML techniques, such as transfer learning and meta-learning, into PINNs to maximize knowledge reuse, reduce adaptation time, and minimize data requirements. In addition, these methods show potential in addressing some of the convergence challenges associated with PINNs. This survey highlights the idea of efficient model adaptivity for PINNs and its potential to facilitate broader adoption in real-world applications where data are scarce and fast evaluation is essential.

The key contributions of this work include:

1. An introductory overview of PINNs, highlighting their connections to traditional numerical methods for solving PDEs and more traditional techniques, such as reduced-order modeling (ROM), that reuse solution data across similar PDE problem instances.

2. A review of recent advances for PINNs, focusing on techniques such as transfer learning and meta-learning to improve model adaptivity and efficiency.

3. Identification of potential metrics and benchmarks to assess the adaptivity of the methods.

4. Future research directions and potential applications of adaptive PINNs across various domains.

The paper is structured as follows. First, section 2 introduced key concepts and terminology. Section 3 provides an overview of how transfer learning and meta-learning enhance the adaptivity of PINNs. Section 4 examines benchmarking methodologies and metrics to assess adaptation efficiency. Section 5 explores real-world applications, discusses limitations and areas for improvement, and outlines future research directions. Finally, Section 6 presents the conclusions.

## 2 Background

### 2.1 Initial Boundary Value Problem

In science and engineering, a problem is often framed as an Initial Boundary Value Problem (IBVP), which encompasses a wide range of phenomena. An IBVP is typically represented as:

$$
\begin{aligned}
\mathcal{N}[u(x,t;\mu)] &= f(x) & \forall \ x \in \Omega \ , \ t \in [t_0, T], \\
\mathcal{B}[u(x,t)] &= g(x) & \forall \ x \in \delta\Omega \ , \ t \in [t_0, T], \\
\mathcal{I}[u(x,t)] &= h(x) & \forall \ x \in \Omega, \ t = t_0,
\end{aligned}
\tag{1}
$$

where, $\mathcal{N}$ represents the differential operator acting on the function $u$, which depends on parameters denoted by $\mu$. $f(x)$ represents the source term defined in the domain $\Omega$. The operator $\mathcal{B}$ imposes the boundary conditions $g(x)$ on $u$ at the boundary $\partial\Omega$ of the domain. Lastly, $\mathcal{I}$ sets the initial conditions $h(x)$ for the function $u(x,t_0)$, representing the initial state of $u$ within $\Omega$ at the initial time $t_0$. The differential operator $\mathcal{N}$ can take the form $\mathcal{N}[u(x,t;\mu)] = F(x, u, \frac{\partial u}{\partial t}, \frac{\partial u}{\partial x}, \frac{\partial^2 u}{\partial x^2}, \ldots)$, where $F$ is some given function that describes the dynamics of the system. The objective in solving an IBVP is to find the function $u(x,t;\mu)$ that satisfies the differential equation, the boundary conditions, and the initial conditions simultaneously.

Three methods are often used to solve such IBVP problems numerically: the Finite Element Method, the Finite Difference Method, and the Finite Volume Method. While these methods differ in their mathematical formulations and approaches, they all discretize the domain into smaller subdomains (elements, cells, or grid points) and perform local approximations to obtain the global solution.

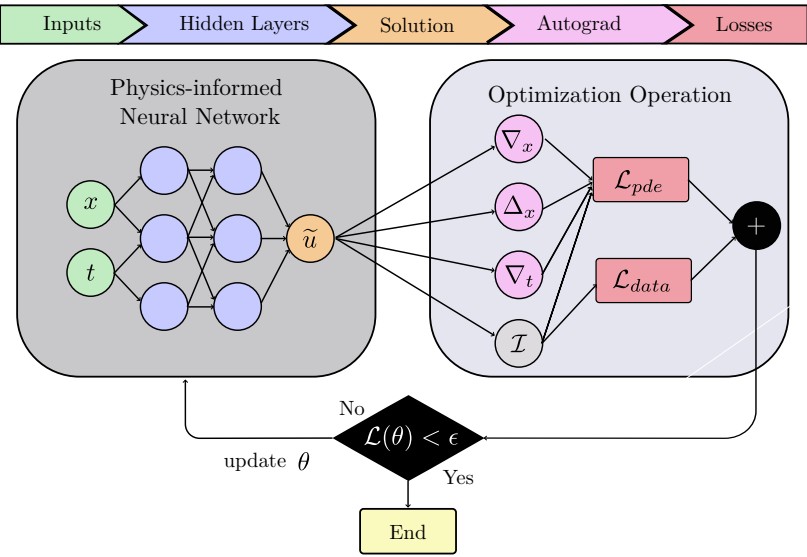

Figure 2: Schematic representation of PINNs: The network is optimized by minimizing a composite loss function that combines a regression loss from observed data, PDE residuals at collocation points, and boundary/initial condition losses.

## 2.2 Physics-Informed Neural Networks

Physics-informed Neural Networks approximate the solution $u$ by using a neural network $u_\theta$ and incorporating IBVP information directly into the training process. Although various methodologies exist, we focus on one of the most common approaches, which involves incorporating the residual of the IBVP into the loss function Raissi et al. (2019).

Considering the definition of the IBVP (1), the equations can be reformulated in terms of their residuals. These residuals are computed at collocation points within the corresponding domain ($\Omega$, $\delta\Omega$, or $[t_0, T]$), sampled at discrete locations denoted $N_{\text{PDE}}$, $N_{\text{BC}}$, and $N_{\text{IC}}$.

$$
\begin{aligned}
\mathcal{N}[u_\theta(x,t;\mu)] - f(x) &= r_{\text{PDE}} & \forall \ x \in \{x_i\}_{i=1}^{N_{\text{PDE}}}, \ t \in \{t_i\}_{i=1}^{N_{\text{PDE}}}, \\
\mathcal{B}[u_\theta(x,t)] - g(x) &= r_{\text{BC}} & \forall \ x \in \{x_i\}_{i=1}^{N_{\text{BC}}}, \ t \in \{t_i\}_{i=1}^{N_{\text{BC}}}, \\
\mathcal{I}[u_\theta(x,t)] - h(x) &= r_{\text{IC}} & \forall \ x \in \{x_i\}_{i=1}^{N_{\text{IC}}}, \ t = t_0.
\end{aligned}
\tag{2}
$$

Here, the derivatives of the differential operator $\mathcal{N}[\cdot]$ are computed using automatic differentiation.

The loss function is defined as a weighted sum of the individual loss terms, where each term is given by $\mathcal{L}_{(.)} = \text{MSE}(r_{(.)})$, with the placeholder (.) representing the PDE residuals, the boundary condition (BC) or the initial condition (IC). The weights $w_{\text{PDE}}$, $w_{\text{BC}}$, and $w_{\text{IC}}$ balance the contribution of each term of the loss

$$
\mathcal{L}(u_\theta) = w_{\text{PDE}} \, L_{\text{PDE}} + w_{\text{BC}} \, L_{\text{BC}} + w_{\text{IC}} \, L_{\text{IC}}.
\tag{3}
$$

By optimizing the network parameters $\boldsymbol{\theta}$ with respect to the loss function, the network $u_\theta$ aims to learn a solution $u$ that approximates the true solution. If partial observation data[3] are available, such as from experiments or sensors, an additional regression loss term can be incorporated into Equation 3.

## 2.3 Weighted Residuals: Collocation Method

To highlight the similarities between PINNs and numerical methods, this paper compares the collocation method with the PINN approach. For simplicity, a steady 1-D case will be considered. The residual of an

---

[3]The terms partial observation data, ground truth data, and sensor data are used interchangeably throughout this work.

IBVP is formulated as:

$$\mathcal{N}[u(x;\mu)] - f(x) = R(x) \qquad \forall\ x \in \Omega \tag{4}$$

An approximate solution $\widetilde{u}$ is devised in such a form that it is possible to approximate a wide range of functions:

$$\widetilde{u}(x) = \sum_{i=1}^{N} a_i \cdot \phi_i(x), \tag{5}$$

where, given a good choice of basis $\boldsymbol{\phi}$, the task is to find the expansion coefficients $\boldsymbol{a}$ such that the residual is minimized. Since the problem is defined over a continuous domain, minimizing the residual requires integrating it over the domain. To account for the spatial variation of the residual, the residual is weighted by $\omega(x)$, leading to the weighted residual formulation:

$$\int_{x_0}^{x_f} \omega(x) \cdot R(x)\ dx = 0. \tag{6}$$

Here, $\omega(x)$ is a weighting (or test) function chosen to enforce the orthogonality of the residual $R(x)$. Common choices include piecewise polynomials, as in finite element methods, or Dirac delta functions, as used in the method described here.

The collocation method is a special case of the weighted residual formulation (6), where $\omega(x)$ is chosen as a Dirac delta function: $\omega(x) = \delta(x - x_i)$. This allows direct evaluation of the residual at specific locations. A system of equations can be constructed to approximate the solution by selectively minimizing the residual at these points. The basis functions $\phi_i(x)$ are selected based on the requirements of the problem and play a crucial role in determining the precision and effectiveness of the approximation. Typically, they are chosen to satisfy the boundary conditions and ensure linear independence. These basis functions should possess properties that allow them to accurately capture the behavior of the solution within the problem domain.

## 2.4 Reduced Order Modeling: A numerical approach for reusing information

In the pursuit of efficient model adaptation, it is valuable to explore numerical methods that leverage previously obtained solutions to infer new similar solutions, thus reducing computational demands. Reduced order modeling (ROM) encompasses a class of numerical techniques that aim to construct a simplified version of the original model by reducing its computational complexity. This is achieved by constructing a low-dimensional approximation that captures the essential behavior of the high-fidelity model or simulation but with significantly fewer degrees of freedom. The key objective of ROM is to enable efficient adaptation to new scenarios by leveraging known information from existing simulations, experimental data, or solutions to similar problems.

One such approach is the Galerkin method combined with Proper Orthogonal Decomposition (POD), often referred to as the Galerkin-POD method. This approach uses the known information from existing solutions (snapshots) to construct a reduced basis consisting of POD basis functions that capture the essential dynamics of the system in various scenarios. This diverse set of snapshots, obtained from multiple tasks or parameter configurations, encapsulates the shared knowledge and dominant features of the system's behavior. By performing Proper Orthogonal Decomposition (POD) on these snapshots, the method extracts the dominant POD basis functions that serve as a compact representation of the solution manifold.

Considering the IBVP problem defined in Section 1, the goal of the Galerkin-POD method is to find an approximate solution $\widetilde{u}(x, t; \mu)$ expressed as a linear combination of the extracted POD basis functions $\phi_i(x)$:

$$\widetilde{u}(x, t; \mu) = \sum_{i=1}^{N} a_i(t; \mu) \cdot \phi_i(x), \tag{7}$$

where $a_i(t; \mu)$ are the time-dependent modal coefficients and $N$ is the number of retained POD basis functions.

The reduced basis is then used to project the governing equations onto a reduced subspace, yielding a reduced system of equations for the modal coefficients. Consequently, when faced with a new task or

scenario, the Galerkin-POD method can efficiently adapt the solution by solving this reduced system. By reducing the dimensionality, the resulting reduced-order model becomes computationally less expensive to solve or simulate while maintaining an acceptable level of accuracy.

## 2.5 Relationship with PINNs

Both PINNs and the collocation method leverage residual information to guide the approximated solution toward the ground-truth solution of the governing equations. However, they differ in their approach to constructing the solution ansatz. PINNs exploit the universal approximation theorem, using neural networks as the ansatz solution, with the ability to capture local behavior and sharp gradients dependent on the choice of the activation function. In contrast, the collocation method builds the solution as a linear combination of linearly independent basis functions and expansion coefficients. The selection of basis functions, whether global or piecewise, involves a trade-off between accuracy and computational efficiency. Piecewise basis functions, with their compact support and ability to capture local behavior, can provide higher accuracy for problems with localized features or complex geometries, but at the cost of increased computational complexity and larger systems of equations. Global basis functions, on the other hand, are generally smoother and require fewer basis functions, leading to smaller systems and simpler implementation, but may struggle to accurately represent sharp gradients or complex geometries.

The POD-Galerkin method takes a different approach by building a global basis that generalizes to several PDE instances, resulting in fewer equations to solve. By projecting the governing equations onto the reduced space spanned by the global basis, the POD-Galerkin method transmits information from other solutions through the basis functions, enabling efficient and accurate approximations for a range of PDE instances.

It is worth noting that the linear combination of basis functions in the collocation and POD-Galerkin methods provides a structured and lower-dimensional representation of the solution space, improving computational efficiency. However, this structured approach inherently limits expressibility compared to the more flexible function approximation enabled by PINNs under the universal approximation theorem. This trade-off has motivated recent efforts to frame PINNs in a similar linear combination form to balance efficiency and expressibility (Chen & Koohy, 2024; Desai et al., 2021; Peng et al., 2020; Penwarden et al., 2023; Bischof & Kraus, 2022).

## 2.6 Efficient Model Adaptivity

Efficient model adaptivity refers to the ability of a machine learning model to efficiently and effectively adjust to new, previously unseen tasks using knowledge gained from previous tasks. Given a model pre-trained on one or multiple source tasks $T_s \subset T$, the goal is to adapt this model to an arbitrary novel target task sampled from $t \sim T_t$ or several target tasks $T_t \subset T$ where $T_t \cap T_s = \emptyset$. It is assumed that all tasks from $T$ share some common characteristics. In the context of PINNs, a model is typically trained per task, where each task corresponds to an instance of the IBVP, each subject to different parameters $\mu$, which can be a material property, boundary condition, or initial condition. Figure 3 illustrates two examples of IBVP. The first example corresponds to the 2D heat equation, where the task-defining parameter is the diffusivity coefficient. The second example corresponds to the Burgers' equation, where the task is defined by its initial condition.

Key aspects that affect efficient model adaptivity include computational efficiency and data efficiency. Computational efficiency refers to the model's ability to adapt quickly with minimal computational resources. It encompasses the speed of adaptation, the amount of processing power required, and the overall time needed to adjust the model for new tasks. Computational efficiency is influenced by several factors, primarily the number of model parameters, the complexity of the model, and the optimization steps required for adaptation. These elements directly impact the overall training time and computational resources needed. On the other hand, data efficiency refers to how effectively a model can learn from limited data samples. This data can encompass various types: collocation points (evaluation points), partial observations, or the pre-training tasks needed for generalization. This work surveys the application of transfer learning and meta-learning to physics-informed neural networks to achieve efficient model adaptation.

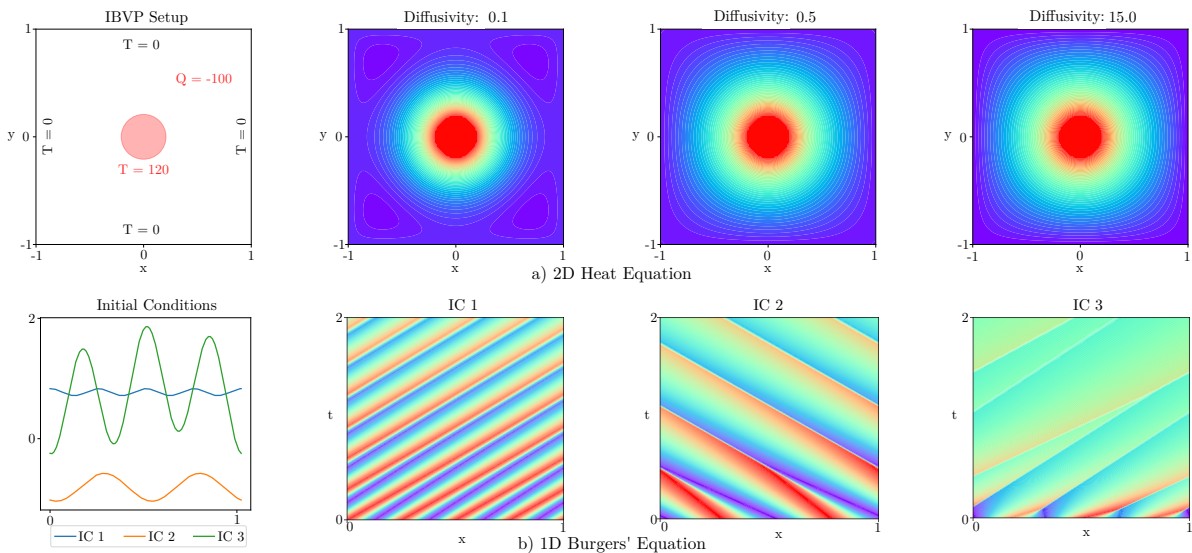

Figure 3: Illustration of IBVP as tasks: a) Heat equation tasks with varying material properties. b) Burgers' equation tasks with different initial conditions (adapted from Takamoto et al. (2022)).

## 2.7 Transfer Learning and Parameter-Efficient Fine-Tuning

Transfer learning is a machine learning approach that transfers knowledge gained from a source domain to a different but related target domain. The fundamental principle is to leverage a model pre-trained on a source task or dataset and adapt its learned representations to a new target task. Transfer learning aims to accelerate the learning process and improve generalization performance compared to training from scratch by fine-tuning the pre-trained model on the target data. This technique is particularly beneficial when the target task has limited labeled data.

Parameter-efficient fine-tuning (PEFT) is an advancement in transfer learning that aims to make the fine-tuning process even more efficient and scalable. Unlike traditional fine-tuning, which updates all the parameters of the pre-trained model, PEFT introduces a small number of trainable parameters that modulate the pre-trained model's behavior. These additional parameters are trained to adapt the pre-trained model to the target task, while most of the original model parameters remain frozen during fine-tuning. Although PEFT was developed for large-scale models, such as large language models, we use the term to describe the selective fine-tuning of smaller models, such as those used in PINNs. This interpretation focuses on the principle of updating only a subset of parameters to achieve efficient adaptation, regardless of model size.

---

### A Brief History of Transfer Learning

Research in transfer learning dates back to the 1970s and 1980s, as described in the work of Bozinovski (2020). Pratt et al. (1991) conducted pioneering studies that explored to what extent neural networks can be "recycled," coining this the "transfer problem." Sharkey & Sharkey (1993) focused on using prior knowledge to improve the performance of new tasks and on understanding when knowledge can be transferred between networks. Sharkey's work suggests that transfer learning has strong roots in psychology. In 1995, the fundamental motivation for transfer learning was discussed in the NIPS-95 workshop on "Learning to Learn" by Baxter et al. (1995), as referenced by Pan & Yang (2009). The first comprehensive survey on transfer learning was published by Pan & Yang (2009). More recently, Zhuang et al. (2020) offered an extensive survey covering more than 40 representative transfer learning approaches from both the data and the model perspective.

---

## 2.8 Meta-learning

Meta-learning is a field in machine learning that encompasses the notion of "learning to learn." The core idea is to leverage an external architecture or algorithm beyond single-task learning models, which can capture the relationships and shared knowledge across different tasks. By learning the correlations between various tasks during a meta-training phase, this external meta-learner can then modify, e.g., the structure of the base learning algorithms, their hyperparameters, and/or the model architectures. This allows the meta-learner to adapt and transfer the acquired meta-knowledge to new, unseen tasks more efficiently, facilitating rapid learning and generalization. This distinguishes itself from traditional transfer learning, which focuses on adapting between a source and target domain. Meta-learning aims to extract task-general knowledge from a distribution of tasks, enabling systematic adaptation to any task within that distribution.

**A Brief History of Meta-learning**

The foundations of meta-learning in machine learning can be traced back to several seminal works. For instance, Schmidhuber (1987) introduced the concept of "self-referential learning," a technique where a network receives its own weights as inputs and predicts new updates for such weights[a]. Bengio et al. (1990) hypothesized that it is possible to learn algorithms for synaptic learning rules and that these rules could be constrained such that the resulting neural networks can solve complex AI tasks. Later, Hochreiter et al. (2001) demonstrated that gradient-based optimization can be leveraged to automatically discover effective algorithms tailored to specific tasks, such as time series forecasting. Expanding on these ideas, Finn et al. (2017) introduced Model-Agnostic Meta-Learning (MAML), a technique designed to enable fast adaptation to new tasks with minimal fine-tuning. MAML employs bi-level optimization: in the inner loop, task-specific parameters are updated through a few gradient steps; in the outer loop, the initialization itself is optimized to minimize the loss across multiple tasks. This approach ensures that the learned initialization is well-suited for rapid adaptation. However, MAML's reliance on second-order derivatives in the outer-loop optimization introduces significant computational overhead. To overcome this limitation, Nichol et al. (2018) introduced Reptile, a simplified first-order approximation to MAML that avoids second-order derivatives. Instead of explicitly computing these gradients, Reptile updates the initialization by directly using the difference between task-specific parameters after inner-loop updates. This approach retains much of MAML's effectiveness while being computationally efficient. These foundational developments have significantly influenced subsequent meta-learning techniques, which continue to play a pivotal role in enabling efficient and adaptable learning across diverse tasks.

---

[a](Hospedales et al., 2021)

# 3 Methods

## 3.1 Transfer Learning in PINNs

Transfer learning has emerged as a valuable technique to enhance the efficiency and scalability of PINNs. By leveraging knowledge from pre-trained models, transfer learning addresses the computational challenges and convergence issues often encountered when training PINNs from scratch. This section reviews key advances in the application of transfer learning to PINNs, with an overview of the literature discussed provided in Table 1.

### 3.1.1 Full Fine-tuning

Full model fine-tuning (FFT) is a transfer learning strategy that adapts a pre-trained model to a new task by updating all its parameters. FFT facilitates efficient adaptation of PINNs, particularly when the source and target tasks are similar. This approach is especially useful in computationally intensive scenarios where training a PINN from scratch for each new task would be prohibitively expensive.

An example of full model fine-tuning in PINNs is the TL-PINN method introduced by Prantikos et al. (2023), which addresses the Point Kinetic Equation[4], a critical tool for real-time reactor analysis. TL-PINN employs transfer learning to accelerate training by pre-training a PINN on a source task, such as simulating reactor behavior under nominal conditions or controlled parameter variations (e.g., temperature shifts). The pre-trained model is then fine-tuned for a target task involving distinct transients, such as different reactivity insertion schedules. TL-PINN has been shown to reduce training iterations by an order of magnitude compared to conventional PINNs. Performance improvements correlate with task similarity, measured using geometric metrics like the Hausdorff distance. This framework provides a practical guideline: TL-PINN significantly improves efficiency without sacrificing accuracy when source and target tasks share dynamical features (e.g., temporal profiles of neutron flux).

Zhou & Mei (2023) proposed a method that combines the smoothed finite element method (S-FEM) with PINNs to address inverse problems, specifically the inversion of material parameters in scarce data regimes. In such regimes, a common strategy is to pre-train on solver data and fine-tune with limited, partial observations. The authors use S-FEM to generate high-quality data for pre-training the PINN, and then, as a proof of concept, they fine-tune the model with additional S-FEM data to infer new parameters and evaluate their approach. S-FEM is preferred over FEM as it generates higher-quality data, which, as demonstrated in their results, enhances the pre-trained model and ultimately improves fine-tuning accuracy. Their findings indicate that the use of transfer learning with PINNs increases computational efficiency by a factor of two compared to standard PINNs. The authors emphasize the importance of addressing negative transfer in transfer learning, noting that the success of transfer learning depends on the assumption that the source and target domains share similarities.

Table 1: Transfer Learning in Physics-informed Neural Networks.

| FS. | Literature | Task | PT | PType | Benchmark Equations |
|---|---|---|---|---|---|
| | **Transfer Learning Strategies in PINNs** | | | | |
| FFT | Prantikos et al. (2023) | ODE | ST | Fwd. | *Point Kinetic (PKEs) |
| | †Zhou & Mei (2023) | PDE | ST | Inv. | *Elastoplastic$^{2D}$ |
| PEFT | Desai et al. (2021) | PDE/ODE | MT | Fwd. | Pois.$^{2D}$, Schr.$^{1D}$, 1st/2nd-order ODEs |
| | Goswami et al. (2020) | PDE | ST | Fwd. | *Fracture Mechanics$^{2D}$ |
| | Gao et al. (2022) | PDE | ST | Fwd. | Linear Parabolic$^{10D}$, Allen Cahn$^{10D}$ |
| | Pellegrin et al. (2022) | ODE | MT | Fwd. | Stochastic Branched Flow$^{2D}$ |
| | †Chakraborty (2021) | PDE/ODE | ST | Fwd. | Stochastic ODE, Burgers$^{1D}$ |
| CTL | Lin & Chen (2024) | PDE | ST | Inv. | Schrödinger$^{1D}$ |
| | †Xu et al. (2022) | PDE | MT | Inv. | *Elastic$^{2\text{-}3D}$, Hyperelastic$^{2D}$ |
| | †Mustajab et al. (2024) | ODE/PDE | ST | Fwd. | Harmonic Oscillator, Wave Equation$^{1D}$ |

**Note:** Fine-tune Strategy (FS), Pre-train type (PT), Problem Type (PType), Full Fine-tune (FFT), Parameter-efficient Fine-tuning (PEFT), Curriculum Transfer Learning (CTL), Single-task Learning (ST), Multi-task Learning (MT). Equations with (*) are domain-specific problems. References marked with (†) indicate the use of few-shot learning techniques. Abbreviations: Poisson = Pois., Schrödinger = Schr.

### 3.1.2 Parameter-Efficient Fine-tuning

PEFT techniques were originally introduced as a method to fine-tune large-scale pre-trained models while minimizing computational and memory requirements. These methods typically introduce a small set of trainable parameters, such as adapters or low-rank updates, allowing for efficient adaptation. In this context, the term PEFT is adapted to describe selective fine-tuning strategies within PINNs that aim to reduce computational costs by limiting the number of updated parameters. Although the scale of PINNs may not

---

[4]The Point Kinetic Equation is a simplified model used to analyze the behavior of nuclear reactors over time (reactor transients). It consists of a system of stiff nonlinear ordinary differential equations that model the kinetics of reactor variables.

align with the large models typically associated with PEFT, the underlying principles of parameter efficiency and adaptability remain relevant. The studies discussed here illustrate how these principles can enhance the scalability and performance of PINNs, particularly in data-scarce or resource-constrained scenarios.

In the context of PINNs, Desai et al. (2021) proposes a pre-training and fine-tuning strategy for PINNs to efficiently solve linear ODEs and linear PDEs. During the pre-training phase, the method learns a set of shared bases, represented as $\phi(x)$, which is derived from the hidden layers $(H^{L-1} \circ H^{L-2} \circ \cdots \circ H^0)(x)$, by training on multiple source tasks involving ODEs or PDEs. During the fine-tuning phase, the focus shifts to determining the expansion coefficients, represented by the weights of the output layer $\alpha = W_{\text{out}}$, for a new problem instance. If $\alpha$ can be obtained analytically in closed form, it is computed by solving a linear system of equations, reducing the computational cost to a matrix inversion. For more complex cases, $\alpha$ is optimized through gradient descent, where only the output layer $\alpha$ is updated, while the shared bases $\phi(x)$ remain frozen. The final solution $\widetilde{u}(x)$ is expressed as:

$$\widetilde{u}_\theta(x) = \alpha \cdot \phi(x). \tag{8}$$

This approach enables efficient transfer of shared bases $\phi(x)$ between tasks and significantly reduces training overhead. The accuracy of the final solution depends on how well the bases $\phi(x)$ cover the solution space of the target problem. By calculating the coefficients $\alpha$ in closed form for linear systems, this method achieves one-step adaptation, allowing the shared bases $\phi(x)$ to be adapted to new problems without iterative training or updates to the hidden layers. Future directions of this work include extending the method to incorporate real-world observational data, expanding the framework to nonlinear PDEs, and exploring the characteristics of shared bases to provide better generalization and adaptivity across tasks.

Goswami et al. (2020) proposes a method for phase-field fracture modeling using PINNs enhanced with transfer learning. Unlike conventional residual-based PINNs, this approach minimizes the variational energy of the system while enforcing boundary conditions as hard constraints. This formulation offers two key advantages: 1) imposing boundary conditions is simpler and more robust, and 2) the resulting equations involve lower-order derivatives, making training faster. The fracture modeling process involves iteratively updating the displacement field $u$, which describes material deformation, and the phase field $\phi$[5] by minimizing the total energy at each small displacement step. To overcome the high computational cost of retraining the PINN at every step, transfer learning is employed: only the last layer's weights and biases are retrained, using the previous step's parameters as initialization. This transfer-learning approach significantly improves computational efficiency compared to standard PINNs by requiring fewer iterations to achieve convergence and substantially reducing the time required for each iteration.

Chakraborty (2021) proposes a transfer learning approach to approximate high-fidelity models using PINNs. The method begins by training a PINN on a low-fidelity model of a given IBVP task. Then, a transfer learning technique is applied, where only the last one or two layers of the network remain trainable. This pre-trained model is subsequently fine-tuned to approximate a higher-fidelity model using limited high-fidelity observations of the same task. The approach is particularly useful in scenarios where the exact high-fidelity model is not known a priori, allowing efficient adaptation to different boundary conditions or initial conditions.

Gao et al. (2022) introduces SVD-PINNs, a transfer learning method designed to solve high-dimensional PDEs efficiently. The core idea involves pre-training a PINN, which consists of a three-layer MLP, on an arbitrary PDE task. During fine-tuning for a new PDE, the weight matrix of the middle layer is decomposed using SVD (Singular Value Decomposition). The singular vectors, which capture intrinsic patterns from the source task, are kept frozen, while the singular values and weights of the initial and final layers are adapted. The authors demonstrated the effectiveness of SVD-PINNs in 10-dimensional linear parabolic equations and Allen-Cahn equations, showing superior performance in terms of relative error and convergence speed compared to vanilla PINNs and other transfer learning methods. A notable advantage of SVD-PINNs is their efficiency in solving multiple related PDEs with identical differential operators but different right-hand side functions. However, the main challenge lies in optimizing the singular values during training. Successful

---

[5]Phase-field modeling tracks fracture using a continuous scalar field $\phi$. This scalar field represents the damage state of the material, smoothly transitioning from intact to fractured material, and models the evolution of cracks without sharp discontinuities.

optimization of these values leads to better performance than methods that freeze the first layer, as seen in previous transfer learning approaches for PINNs. In contrast, biased or inaccurate singular values can result in worse outcomes than prior methods.

Pellegrin et al. (2022) presents a multi-task learning strategy aimed at improving training efficiency and performance by leveraging knowledge from related tasks. Their approach employs a multi-head architecture with two primary phases: pre-training and fine-tuning. Initially, during the pre-training phase, the multi-head model, comprising a shared base neural network along with multiple task-specific heads, is trained concurrently on various related PDE tasks. Through training on these interrelated tasks concurrently, the shared base network learns to extract relevant features and representations that encapsulate the underlying dynamics common across the tasks. In the subsequent fine-tuning phase, the weights of the pre-trained base network are fixed, and a new task-specific head is introduced and fine-tuned for the target transfer learning task. This fine-tuning phase enables the model to adjust the learned shared representations to the specific requirements of the new task while leveraging the knowledge gained from related tasks during pre-training. By leveraging the shared base network learned from multiple related tasks, the model can potentially converge faster and achieve better performance on the target task than training a PINN from scratch.

### 3.1.3 Curriculum Transfer Learning

Curriculum Transfer Learning involves progressively increasing the complexity of tasks during the transfer process, starting with simpler tasks and moving towards more complex ones. An example of this approach is the work of Mustajab et al. (2024), who employed a curriculum transfer learning strategy to address high-frequency and multi-scale problems. The method begins by training on relatively simple low-frequency problems, which are easier to solve, and then gradually escalates to more challenging high-frequency tasks, transferring knowledge gained from the lower-frequency problems. Through full-weight transfer learning, this strategy successfully enabled the model to learn high-frequency solutions that traditional PINNs cannot achieve without increasing the number of layers. However, the authors highlight the importance of understanding the limitations of transfer learning, particularly in terms of when it succeeds and when it may fail. Overall, this approach enhances the convergence speed and robustness of PINNs for high-frequency and multi-scale PDEs.

Gradient-enhanced PINNs (gPINNs), introduced by Yu et al. (2022), improve standard PINNs by incorporating the gradient of the residual as an auxiliary loss term. This approach captures local information around collocation points, enhancing accuracy but increasing computational overhead due to added optimization complexity. To address this limitation, Lin & Chen (2024) proposes TL-gPINNs, an extension of gPINNs tailored for inverse problems with variable coefficients, such as time-dependent boundary conditions. TL-gPINNs adopt a two-step optimization strategy: they first train a standard PINN on a simplified objective and then fine-tune the whole model using the gPINN loss. This approach reduces both error and computational cost compared to training gPINNs from scratch. In particular, experiments demonstrated that for PDEs involving multiple loss terms, standard PINNs can sometimes outperform gPINNs, as the additional loss terms increase the optimization complexity. By progressively introducing complexity through the gPINN objective, TL-gPINNs employ a curriculum learning framework to balance computational efficiency and solution accuracy.

Xu et al. (2022) addressed the challenge of inverse analysis in engineering structures, where acquiring data for structural components is often expensive. To this end, the authors sought to improve the training efficiency and accuracy of PINNs for inverse problems through a multi-task transfer learning approach. Their proposed solution involves a two-stage learning process. Initially, in the pre-training stage, a simplified loading scenario is used to pre-train the PINN model, including both the model weights and task-independent loss balancing weights. Subsequently, in the fine-tuning stage, the pre-trained model is partially fine-tuned with data from real engineering problems, with only the last two layers updated. The method was applied to 2D linear and hyperelasticity problems in solid mechanics. Combining layer freezing with inherited multi-task weights from pre-trained models significantly accelerated training convergence.

### 3.2 Meta-learning in PINNs

The integration of meta-learning techniques with physics-informed neural networks has shown promise in improving model adaptivity and generalization. Table 2 presents the taxonomy used to study these techniques in the context of PINNs, focusing on the meta-learning representation ("what is being meta-learned") (Hospedales et al., 2021). In the following section, various studies are introduced according to this taxonomy.

Table 2: Meta-learning Strategies in Physics-informed Neural Networks.

| Meta-learning Strategies in PINNs | | | | |
|---|---|---|---|---|
| **Type** | **Approach** | **Literature** | **PType** | **Equation** |
| Weight Init. | FFT | Liu et al. (2022) | Both | Pois.$^{1\text{-}2D}$, [Burg., Schr.]$^{1D}$ |
| | | Zhong et al. (2023) | Fwd. | *Plasma Sim.$^{1D}$ |
| | | Penwarden et al. (2023) | Fwd. | [Burg., Heat]$^{1D}$, [A-C, D-R]$^{2D}$ |
| | | †Cheng & Alkhalifah (2024) | Fwd. | Wavefield$^{2D}$ |
| | | Qin et al. (2022) | Fwd. | Burg.$^{1D}$, [Pois., Hyp.-elast.]$^{2D}$ |
| | PEFT | Cho et al. (2024b) | Fwd. | C-D-R$^{1D}$, Helm.$^{2D}$ |
| Net. Struct. | Lay./Neur. | Chen et al. (2021) | Inv. | A-D-R$^{1D}$ |
| | Activations | Bischof & Kraus (2022) | Fwd. | Pois.$^{2D}$ |
| | | †Chen & Koohy (2024) | Fwd. | [Burg., K-G, A-C]$^{1D}$ |
| Input | Sampling Points/Params | †Toloubidokhti et al. (2023) | Fwd. | [Burg., Conv., R-D]$^{1D}$, Helm.$^{2D}$ |
| | | Tang et al. (2023) | Fwd. | Ellip.$^{2\text{-}10D}$, Nonlinear PDE$^{10D}$ |
| | Latent Rep. | Huang et al. (2022) | Fwd. | Burg.$^{1D}$, [Max., Laplace.]$^{2D}$ |
| | | Iwata et al. (2023) | Fwd. | Arbitrary Param. PDE$^{1D}$ |
| Loss | Param. Loss | Psaros et al. (2022) | Both | [Adv., Burg.]$^{1D}$, SS R-D$^{2D}$ |
| | Loss Attention | Song et al. (2024) | Fwd. | Burg.$^{1D}$, [LDC Flow, Pois.]$^{2D}$ |

**Note:** Problem Type (PType), Forward Problem (Fwd.), Inverse Problem (Inv.). Equations marked with (*) represent domain-specific problems. References marked with (†) indicate the use of few-shot learning techniques. Abbreviations: Poisson = Pois., Burgers = Burg., Schrödinger = Schr., Simulation = Sim., Allen-Cahn = A-C, Diffusion-Reaction = D-R, Convection-Diffusion-Reaction = C-D-R, Helmholtz = Helm., Advection-Diffusion-Reaction = A-D-R, Klein-Gordon = K-G, Reaction-Diffusion = R-D, Elliptic = Ellip., Hyper-elasticity = Hyp.-elast., Maxwell = Max., Parametric = Param., Advection = Adv., Steady State = SS, Lid-driven Cavity = LDC.

#### 3.2.1 Learning the Weight Initialization

Starting from a good weight initialization can lead to faster convergence, better accuracy, and reduced computational costs—a key factor for real-time applications. A well-informed initialization not only reduces training time but also improves generalization, making it a central focus in many studies. Meta-learning for weight initialization, building on transfer learning principles, extends this by learning from the training process across multiple tasks. Techniques such as MAML, Reptile, and hypernetworks identify or generate optimal starting weights, enhancing efficiency and adaptivity. Using these approaches, PINNs can achieve scalable, resource-efficient performance across diverse problem domains. This section reviews these strategies and their impact on advancing the effectiveness of PINNs.

Liu et al. (2022) employs the Reptile algorithm (Nichol et al., 2018) to find a good initialization of the parameters and compares it against other initialization techniques, such as Xavier initialization, in unsupervised, supervised, and semi-supervised settings for the forward and inverse problems. In their experiments, their Reptile weight initialization outperformed Xavier initialization, with the unsupervised Reptile approach performing the best. The authors point out that this initialization can be used with other PINN architectures

and that, in addition to finding a good initialization of the network parameters, this method could serve to provide good starting points for works that use adaptive activation functions or losses.

Zhong et al. (2023) and Cheng & Alkhalifah (2024) both adapted the MAML approach by Finn et al. (2017) to PINNs, aiming to reduce the training steps required for new tasks by leveraging a meta-network for weight initialization. However, their implementations and benchmark problems differ slightly. Both approaches utilize a support set in the inner loop to update the network for a specific task and a query set in the outer loop to optimize the weight initialization. The key distinction lies in how these sets are constructed: Zhong et al. (2023) uses the same PDE task for both the support and query sets, evaluated at different collocation points, while Cheng & Alkhalifah (2024) uses different PDE tasks for the two sets. In terms of benchmark problems, Zhong et al. (2023) applied their approach to plasma simulations, demonstrating that generalization performance strongly depends on the similarity between source and target tasks. In contrast, Cheng & Alkhalifah (2024) focused on seismic wave equations, showcasing faster convergence and higher prediction accuracy compared to vanilla PINNs. However, both methods rely on an initial meta-training phase to establish robust network parameters. This phase involves two gradient computations, one for the inner task and one for meta-initialization, making the process memory-intensive and computationally costly. Cheng & Alkhalifah (2024) further emphasizes the importance of task diversity during meta-learning, finding that incorporating more tasks helped capture a broader range of distribution features, leading to improved generalization. They also observed that using 20 iterations in the inner loop yielded better performance compared to fewer iterations. Finally, they highlighted the potential for combining their approach with other PINN architectures to further enhance adaptability and efficiency.

Qin et al. (2022) compared MAML and LEAP (Flennerhag et al., 2018) weight initialization with PINNs, extending tasks to different geometries and boundary conditions. LEAP, similar to MAML and Reptile, is a general meta-learning framework. However, instead of finding a good initialization based solely on the final state of the weights from different tasks, LEAP considers the entire optimization path. The objective is to minimize the expected length of the path traveled during the training process, allowing more efficient knowledge transfer between learning processes. In their work, they found that MAML outperforms LEAP in accuracy for a given runtime and requires less hyperparameter tuning. However, LEAP's meta-training is faster and less memory intensive.

Penwarden et al. (2023) compared different weight initialization strategies to improve the optimization of PINNs in terms of both time and accuracy. The methods included random initialization, MAML, center initialization, and initialization via interpolation of pre-trained PINN weights. Center initialization involves starting with the pre-trained weights at the center of the parameter space, assuming that this central position is a reasonable initial point for optimization. For the interpolation method, multiple pre-trained PINN weights were interpolated to infer a general weight initialization that would lead to a more accurate and faster convergence. The conclusions of this work were that the interpolation methods provided good initial weights, enhancing the optimization performance in terms of accuracy and time. However, no definitive conclusion could be drawn regarding the superior interpolation method, as performance varied across different PDEs between Spline, RBF (cubic, Gaussian, and multiquadratic), and polynomial interpolation. One surprising finding was that initializing the weights with those of a pre-trained PINN, trained at the center of the parameter space, produced results comparable to the interpolation methods for higher-dimensional PDEs and, for the 1D-task, outperformed MAML. The authors note that their approach assumes that the parameter space is well-behaved, meaning it does not change drastically between parameters, and emphasize that for future work, identifying the boundaries of these regions is important.

Cho et al. (2024b) developed the Hyper-Low-Rank PINN, which combines meta-learning with PEFT to address parametric PDEs more efficiently. The method features a two-phase training process. In the pre-training phase, the weights of the hidden layers of the base model are constructed using an SVD approach, $W = U\Sigma V$, where $\Sigma$ (the singular values) are provided by the meta-network and the singular vectors $U$ and $V$ are part of the base model. The first and last layers of the base model are kept as standard linear layers. During the fine-tuning phase, the meta-network generates adaptive weights for new tasks and trims less significant weights to maintain a compact, hyper-low-rank structure. In addition, the first and last layers are optimized along with the adaptive weights. The authors conclude that their method improves efficiency and accuracy by leveraging meta-learning and low-rank approximations, reducing computational and memory

costs. It effectively handles varying PDE parameters, mitigates failure modes, and outperforms standard
PINNs. According to the authors, future work should focus on extending the framework to handle more
general settings, such as parameters that define initial / boundary conditions or different domain shapes,
and improving the expressiveness of the bases for greater adaptability across a broader range of PDEs.

### 3.2.2 Learning the Network Structure

Learning the network structure involves tailoring the architecture of neural networks, such as the number
of layers, hidden dimensions, and activation functions, to specific tasks or domains of problems. In the
context of PINNs, meta-learning strategies have been employed to dynamically adapt the network structure
to the unique requirements of different PDE tasks, enhancing the convergence and performance of PINNs,
as demonstrated in the following works.

Chen et al. (2021) developed a method to solve the mixed problem (forward/inverse) of the advection-
diffusion-reaction (ADR) system using sparse measurements. Given the stochastic nature of the problem,
they utilized a PINN architecture called sPINN (Zhang et al., 2019). This composite network comprises
several sub-networks, each corresponding to different fluid flow frequencies (modes). Due to the complexity of
determining the optimal number of layers and neurons for each sub-network, Meta-Bayesian optimization was
employed to automate the selection process. The study focuses on realistic scenarios with limited or sparse
data, addressing both forward and inverse problems common in engineering, where some material properties
and sensor measurements are known. According to the authors, future work should address uncertainty
quantification and explore other optimization methods, such as genetic algorithms, greedy methods, or
reinforcement learning, to improve NN architecture.

Bischof & Kraus (2022) proposed a method that combines Mixture-of-Experts (MoE) and PINNs to solve a
single task. In this approach, multiple expert PINNs are trained on different partitions of the input space,
and a gating network learns the optimal weighting of their predictions. This allows experts to specialize in
different regions of the input space, potentially improving overall accuracy. The meta-learning aspect lies
in modulating the gating mechanism to balance the contribution of each expert PINN, thereby enhancing
the accuracy and convergence of the overall model. Some key remarks of this work are that, when different
experts have different architectures, the gating mechanism discarded the networks with tanh activation and
favored sine activation. Another takeaway is that regularization that weights the importance of each expert
is crucial; in this case, the optimal solution was achieved with three learners. With this method, the focus
was on improving convergence and accuracy. This study indicates that future research should be concerned
with increasing performance, efficiency, robustness, and scalability.

Chen & Koohy (2024) introduced GPT-PINN, which combines meta-learning with a task sampling strategy
that dynamically expands a shared basis dictionary. This method aims to solve new parameter instances
$u(x,t;\mu)$ by approximating them as a weighted sum of pre-trained PINNs:

$$u(\boldsymbol{x},\boldsymbol{t};\mu) \approx \sum_{i=1}^{n} \alpha_i(\mu)\phi_{NN}^{\theta^i}(x,t)$$

Here, $\phi_{NN}^{\theta^i}$ represents pre-trained PINNs at different parameter configurations, and $\alpha_i(\mu)$ are parameter-
dependent coefficients learned by a meta-network. The meta-network modulates the influence of each pre-
trained PINN basis $\phi_{NN}^{\theta^i}$ through the coefficients $\alpha_i(\mu)$ for a given PDE instance. If the approximation fails
to meet the accuracy criterion, a new full PINN is trained individually for that parameter instance, and its
solution is added to the set of basis functions $\phi_{NN}^{\theta^i}$, thus continuously expanding the generalization range of
the overall structure. The approach enables efficient adaptation to new parameter instances while continu-
ously improving the model's generalization capabilities by expanding the shared basis dictionary, ultimately
reducing the computational cost and enhancing the performance of PINNs across diverse parameter spaces.

### 3.2.3 Learning the Loss Function

Meta-learning techniques have also been used to optimize loss functions for PINNs, offering an approach to
significantly improve their performance. These strategies go beyond traditional fixed loss formulations by

adaptively learning loss functions tailored to the specific needs of diverse PDE tasks. By dynamically adjusting the weighting of errors, meta-learning enables better handling of challenging problem regions, improving convergence and enhancing the generalization capabilities of PINNs. The following works demonstrate how these meta-learning strategies have been successfully applied to discover effective loss functions and improve PINN performance across a variety of problem domains.

Psaros et al. (2022) proposed a gradient-based meta-learning algorithm that discovers effective loss functions for various PDE problems. The algorithm operates in two phases: meta-training and meta-testing. During the meta-training phase, a parametric loss function is optimized over a distribution of PDE tasks, learning the optimal parameter values that generalize well across the task distribution. In the subsequent meta-testing phase, the meta-learned loss function is employed to train PINNs on unseen PDE tasks that may differ from the original task distribution and PINN architectures used during meta-training. This meta-learning approach significantly improves the generalization performance of PINNs, enabling them to achieve high accuracy even on out-of-distribution scenarios and PINN architectures that were not encountered during meta-training.

Another recent work that uses meta-learning for the loss function is the Loss-Attentional PINN (LA-PINN) by Song et al. (2024). LA-PINN treats the loss function as a learnable component employing multiple loss-attentional networks (LANs) that are adversarially trained alongside the main PINN model. While the main network minimizes the loss via gradient descent, the LANs use gradient ascent to meta-learn point-wise weights for the loss terms, effectively discovering an "attentional function" to distribute different weights to each collocation point error. This loss-attentional meta-learning framework allows tailoring the loss function per problem by leveraging experience from related tasks, potentially enhancing PINN performance over fixed hand-crafted losses. The adversarial training process, inspired by Generative Adversarial Networks (GANs), enables the LANs to assign higher weights to stiff or hard-to-fit regions, aiding convergence by emphasizing the challenging areas during optimization.

### 3.2.4 Learning the input

Learning the input involves adapting the input data provided to neural networks. In the context of Physics-Informed Neural Networks, this can refer to selecting and adjusting the number and locations of collocation points. Additionally, during a two-phase training process, another strategy is to adapt the task sampling strategy in the pre-training phase. A third approach involves self-referential learning, where the meta-learner dynamically adapts parts of the input based on the specific task requirements.

Toloubidokhti et al. (2023) indicates that many existing meta-learning strategies neglect the varying difficulty levels across different tasks and propose that depending on the difficulty, different collocation point positions and densities should be employed accordingly. To address this, they developed a Difficulty-Aware Task Sampler (DATS) for meta-learning of PINNs, which aims to optimize the task sampling probabilities during meta-training to minimize the average performance across all tasks during meta-validation. DATS employs two strategies: adaptively weighting PINN tasks based on their difficulty, allowing more challenging tasks to contribute more to the meta-learning process, and dynamically allocating the optimal number of residual points (collocation points) across tasks, ensuring that more difficult tasks receive a higher collocation point budget. The evaluation of DATS against uniform and self-paced task-sampling baselines on two meta-PINN models across four benchmark PDEs demonstrates that it improves the accuracy of meta-learned PINN solutions and reduces performance disparity among different tasks while using only a fraction of the residual sampling budget required by the baseline methods.

Another significant contribution is made by Tang et al. (2023), who proposed a Deep Adaptive Sampling (DAS) method for solving high-dimensional PDEs using PINNs. The key innovation is treating the residual of the PINN as a probability density function, approximated by a deep generative model called KRnet. The DAS method involves two main components: solving the PDE by minimizing the residual loss on the current set of collocation points and adaptive sampling, where new collocation points are sampled from the KRnet distribution. This method places more points in regions with high residuals, similar to classical adaptive methods like adaptive finite elements. By iteratively refining the training set, the PINN retrains on an updated set with a focus on high-error regions. The KRnet model is meta-learned to approximate

the residual distribution, enabling adaptive sampling tailored to the current PINN solution. The approach is particularly effective for low regularity and high-dimensional PDEs, where uniform sampling is inefficient. The authors provide a theoretical analysis showing the DAS method can reduce error bounds, with numerical experiments demonstrating significant accuracy improvements compared to uniform sampling.

Self-referential meta-learning approaches have also been combined with PINNs. The Meta-Auto-Decoder (MAD) introduced by Huang et al. (2022) utilizes a self-referential approach to learn parametric PDEs. This approach involves pre-training a neural network $u_\theta(x, z)$ to approximate parametric PDE solutions using a physics-informed loss, where $z$ is a tunable input latent vector that implicitly encodes PDE parameters $\mu$. After pre-training, the network is fine-tuned for new PDEs by either fixing weights $\theta$ and tuning $z$ or tuning both $z$ and $\theta$, allowing it to search for solutions on or near the learned solution manifold.

Instead of learning the latent representation implicitly within the same PINN network, Iwata et al. (2023) proposed to leverage multiple meta-networks to encode the governing equations and boundary conditions into a latent vector $z$. The fine-tuning strategy involves initializing the latent vector $z$ with the help of the meta-networks while keeping these meta-networks frozen. The initialized latent vector $z$ is then set to be tunable, and both the PINN weights $\theta$, and the latent vector $z$ are fine-tuned for the new PDE task. In both MAD and Iwata et al. (2023), the PINN weights are initialized according to the final pre-training phase.

### 3.3 Few-shot Learning in PINNs

Few-shot learning is a machine learning paradigm that enables models to generalize effectively from very limited training examples, emphasizing data efficiency and adaptivity to new tasks or domains with minimal supervision. In the context of PINN adaptivity, this involves addressing three key aspects: the number of pre-training tasks, the number of collocation points required for training, and, when regression data is used, its reduction—where the data may include available observations, sensor measurements, or ground truth data. By strategically selecting and sampling tasks, the model can achieve optimal performance during fine-tuning with fewer samples, thereby enhancing its ability to generalize across tasks. Another approach focuses on optimizing the spatial or temporal distribution of collocation points, reducing computational costs while maintaining reliable predictions in under-sampled regions. Additionally, in semi-supervised or supervised settings, methods leverage sparse observations (e.g., sensor data) to reconstruct full solution fields or solve inverse problems, making them particularly suitable for real-world applications where data is inherently scarce. This section highlights works that integrate meta-learning and transfer learning (marked with the "†" symbol in Tables 1 and 2) to achieve few-shot learning, prioritizing adaptable, generalizable, and data-efficient methods for solving diverse PDE problems.

For example, Chen & Koohy (2024) focuses on reducing pre-training samples by gradually selecting tasks with the highest residuals, thus optimizing task sampling for improved performance during fine-tuning. Similarly, Cheng & Alkhalifah (2024) investigates the impact of pre-training sample sizes, finding that increasing the number of samples improves performance, likely due to the greater diversity of features captured in larger datasets.

On the other hand, Toloubidokhti et al. (2023) develops a technique that not only optimizes task sampling based on task difficulty but also dynamically allocates collocation points according to the complexity of the task. This dual approach improves the efficiency of meta-learning for PINNs by addressing task sampling and collocation point allocation.

Finally, high-frequency problems often require a denser distribution of collocation points. In the work of Mustajab et al. (2024), the authors demonstrate that training with a curriculum—starting from a low-frequency problem and gradually progressing to a high-frequency problem—enables the final model to successfully learn the high-frequency problem without increasing the number of collocation points. This highlights how curriculum-based approaches can reduce the need for high-density collocation points, benefiting few-shot learning scenarios.

Building on the goal of reducing data requirements in real-world applications, several studies have developed strategies to use sparse observations effectively. These methods aim to reconstruct solutions or tackle inverse problems with limited supervision. The following studies are particularly relevant in this context:

Zhou & Mei (2023) explored whether a PINN could be pre-trained using a small dataset generated by a solver and then fine-tuned with limited observational data to solve inverse problems, which is a key aspect of few-shot learning. They used solver-generated data during the fine-tuning phase rather than relying on real-world observations. Their work highlights the potential of transfer learning in data-scarce scenarios, offering valuable insights for applications with limited data availability and minimal supervision.

Chakraborty (2021) frames the approach around the assumption that field data is scarce in practice. To address this, the work blends concepts from PINNs and data-driven learning. The primary idea is to first train a low-fidelity model based on the PINN loss and then apply transfer learning to update the model using high-fidelity data. With this approach, the pre-trained model is trained without data, while fine-tuning is performed with only a few data samples.

Similarly, Xu et al. (2022) addressed the challenge of limited data availability by employing multitask pre-training on simplified tasks solvable with numerical solvers, effectively implementing a few-shot learning approach. This strategy enabled the model to leverage abundant and low-cost data for the fine-tuning phase, thereby reducing the number of samples needed for real-world inverse problems with sparse observational data. The study underscores the potential of transfer learning to significantly minimize sample requirements, a core principle of few-shot learning.

Though some of these studies did not explicitly benchmark few-shot learning, they collectively demonstrate how transfer learning and meta-learning can address limited-data problems, ultimately reducing the number of samples required for model training and fine-tuning. This highlights the potential of these approaches to enable efficient learning in scenarios with scarce data, a central tenet of few-shot learning.

## 4 Metrics & Benchmarks

This section introduces the benchmarks and metrics relevant to efficient model adaptation. Section 4.1 presents the common benchmark PDE problems used throughout the works herby surveyed, Section 4.2 outlines error quantification methods for single-task and multi-task scenarios, and Section 4.3 presents key metrics for assessing efficient adaptivity, crucial for evaluating model performance.

### 4.1 Benchmark PDE Problems

Several methods evaluate their performance using a variety of benchmark problems to ensure robustness and reliability. Tables 1 and 2 compile the specific equations adopted in individual studies, providing a comprehensive overview of the problem domains explored. Building on this, Table 4 (available in Appendix A.1) synthesizes the broader landscape of benchmark usage, categorizing equations by their complexity, key drivers of difficulty and prevalence in the literature. This table serves as a practical guide for selecting appropriate benchmarks based on specific requirements for complexity and solution characteristics.

The most commonly used equations in this survey are Burgers' equation (featured in 9 studies) and the Poisson equation (5 studies), which represent intermediate and baseline levels of complexity, respectively. Less frequently used benchmarks include the Allen-Cahn and Schrödinger equations (each used in 3 studies), while the A-D-R and Navier-Stokes equations, due to their higher complexity, are each utilized only once. The popularity of Burgers' equation can be attributed to its balanced combination of manageable computational demands and challenging nonlinear dynamics, making it particularly well-suited for evaluating time-dependent models.

### 4.2 Error Quantification in PINNs

The quantification of errors in PINNs has traditionally relied on the relative $L_2$ error, which measures deviations from ground-truth solutions but does not assess generalization across tasks. While effective in single-task scenarios, multi-task settings require a focus on task similarity—a key factor in evaluating interpolation and extrapolation within the parametric space, as well as generalization through knowledge transfer. To address this, Chen et al. (2021) presented the errors in Table 3, incorporating not only single-task errors and losses but also the 'worst-case' error, which captures the largest error within a set of tasks.

Additionally, they visualize results as cross-task distributions, such as candlestick plots, providing a more comprehensive view of error variation. Optimizing with respect to the 'worst-case' loss further aids training by reducing worst-case errors and enhancing generalization in multi-task scenarios.

A central challenge in benchmarking task similarity lies in rigorously quantifying it. Current methods exhibit key trade-offs. For instance, Huang et al. (2022) pre-train models on PDE parameters drawn from divergent distributions, but this heuristic does not guarantee task dissimilarity, as differing PDE parameters may still result in similar solutions. In contrast, Prantikos et al. (2023) uses geometric metrics such as the Hausdorff distance to measure the dissimilarity between solutions across tasks. This approach involves comparing the solutions of a set of tasks to identify OOD tasks, which are primarily used for benchmarking purposes. However, this method requires finalized solutions, limiting its applicability to scenarios where solutions are already available. Alternatively, adaptive frameworks, such as GPT-PINN (Chen & Koohy, 2024), bypass explicit similarity metrics by identifying OOD tasks based on the initial task-specific loss and retraining models when this loss exceeds predefined thresholds, prioritizing practicality over formal guarantees.

These insights emphasize the importance of integrating task similarity, error distributions, and extreme-value analysis into error assessment frameworks. Such integration is essential for robust model development and for gaining a deeper understanding of generalization, particularly in multi-task settings where knowledge transfer and interpolation/extrapolation within the parametric space are key considerations.

Table 3: Evaluation Metrics for Multi-Task PINNs. "Worst-case" metrics compute the maximum over tasks (parameterized by $\mu$), while "Task-wise" metrics are evaluated per task at the final iteration.

| Metric | Description | Equation |
|--------|-------------|----------|
| Worst-case Loss | Maximum loss across tasks | $\max_{\mu \in \mathcal{T}} \mathcal{L}(\mathbf{u}_\theta; \mu)$ |
| Terminal Loss | Task-specific loss at final iteration | $\mathcal{L}(\mathbf{u}_\theta; \mu)$ |
| Worst-case Rel. $L_2$ Error | Maximum relative error across tasks | $\max_{\mu \in \mathcal{T}} \dfrac{\|\mathbf{u}_\theta(\mu) - \mathbf{u}_{\text{gt}}(\mu)\|_2}{\|\mathbf{u}_{\text{gt}}(\mu)\|_2}$ |
| Terminal Rel. $L_2$ Error | Task-specific relative error at final iteration | $\dfrac{\|\mathbf{u}_\theta(\mu) - \mathbf{u}_{\text{gt}}(\mu)\|_2}{\|\mathbf{u}_{\text{gt}}(\mu)\|_2}$ |
| Terminal Abs. Error | Task-specific absolute error at final iteration | $\|\mathbf{u}_\theta(\mu) - \mathbf{u}_{\text{gt}}(\mu)\|_1$ |

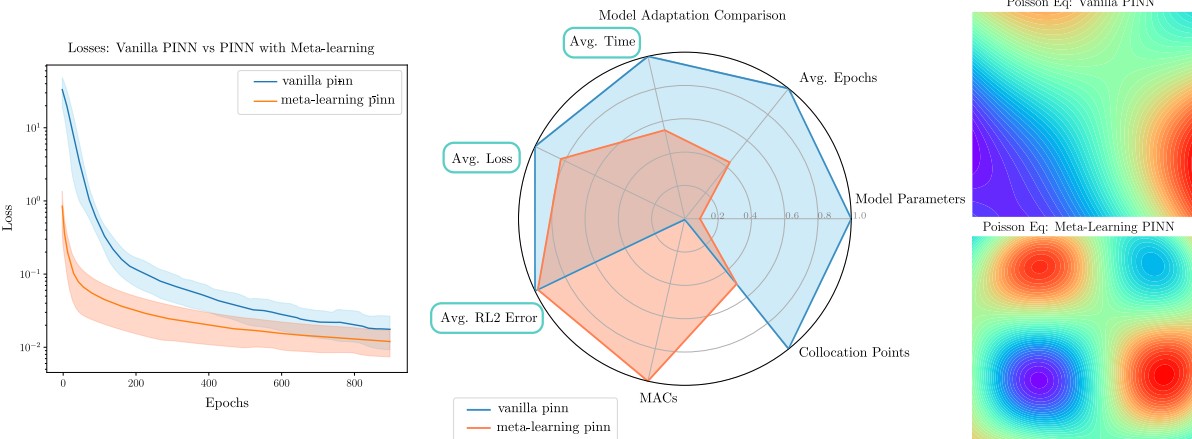

Figure 4: Example of efficient model adaptation through meta-learning. The blue-circled metrics represent key factors that efficient adaptivity aims to reduce, influenced by the other metrics in the radar chart. On the far right is the predicted solution after only 100 epochs.

### 4.3 Efficient Adaptivity Metrics

Evaluating the adaptivity and efficiency of PINNs is crucial for practical applications. Key metrics assess data requirements and computational efficiency, focusing on minimal data usage and reduced training time. Figure 4 compares a vanilla PINN, trained from scratch for each task, with a meta-learning PINN that employs a hypernetwork-based adaptation strategy to leverage prior knowledge (Cho et al., 2024b). The radar chart provides a normalized comparison of key factors influencing efficient adaptation, including the number of epochs, model parameters, collocation points, model complexity, and end metrics such as average convergence time, final loss, and final error. The reported values were obtained by training the models until either a target loss of 0.05 or a maximum of 1,200 epochs was reached. The results demonstrate that meta-learning significantly reduces training time, epochs, and collocation points, thereby lowering computational overhead.

#### 4.3.1 Data Efficiency

**Collocation Point Budget**. The number of collocation points significantly impacts the convergence speed and accuracy of PINNs. Some studies use a fixed number of collocation points, while others employ adaptive sampling strategies, allocating more points to regions with higher PDE losses (Toloubidokhti et al., 2023). A useful evaluation strategy is to measure the accuracy of different sampling techniques given the varying budgets of the collocation points (Wu et al., 2023). By examining the trade-off between the number of collocation points and the resulting accuracy, it is possible to optimize the sampling strategy to suit specific problems and computational constraints, enhancing the training performance and accuracy of PINNs.

**Observation Point Budget**. In many applications, only a limited number of observation points are available for analysis. Rather than relying solely on unsupervised loss, it is important to make the most of these scarce observations. To address this, a metric is designed to assess the relationship between the number of observation points and the accuracy of the analysis. Specifically, it evaluates the accuracy that can be achieved given a fixed budget of $N$ observation points. This is particularly crucial for real-world scenarios where data availability is constrained.

**Task Sampling Strategy**. Several PINN techniques employ a two-phase training process consisting of a pre-training phase and a fine-tuning phase. The pre-training phase can be conducted using either a single instance or multiple instances of PDEs. To achieve better generalization across a distribution of tasks, it is crucial to pre-train on multiple tasks distributed along a given parameter range. However, determining the optimal sampling strategy for selecting pre-training tasks is a challenging task. The goal is to identify the fewest number of pre-training tasks that yield the best results, as the number of pre-training tasks influences both the pre-training resources and the final fine-tuning accuracy. Recent works, such as that of Toloubidokhti et al. (2023), have attempted to address this problem. One approach is to measure the "performance disparity" within a given range of tasks, defined as the performance difference between the worst-performing and the best-performing PINN. If a network architecture generalizes well across the range of tasks, both the accuracy and performance disparity should be low. This analysis can also serve as a tool to assess which PDE tasks an architecture struggles with, providing valuable insights for further improvements.

#### 4.3.2 Computational Efficiency

To evaluate computational efficiency, four key metrics are commonly reported. First, the parameter count provides a measure of the model size, which impacts memory usage. Second, the number of MACs (Multiply-Accumulate Operations) directly reflects the computational complexity, influencing processing speed. Third, the epoch count assesses convergence by reporting either the final accuracy within a set epoch budget or the number of epochs required to reach a target error threshold. Fourth, training time offers a direct quantification of computational cost by measuring the duration needed to achieve the desired accuracy. These metrics collectively provide a comprehensive view of the computational demands associated with different PINN architectures and training strategies, facilitating informed decisions for their deployment in resource-constrained environments.

# 5 Applications & Discussions

Transfer learning and meta-learning techniques have shown considerable promise in enhancing the adaptivity of PINNs. These techniques offer promising solutions for both forward and inverse problems, extending beyond traditional benchmarks to real-life applications in various engineering and scientific domains. This trend highlights how adaptive PINNs are increasingly bridging theoretical advancements with practical utility, as evidenced by the works marked (*) in Tables 1 and 2.

### Adaptivity in Forward Problems

In forward problems, efficient adaptivity focuses primarily on adaptation speed and retrieval of previously unseen solutions from similar tasks. This ability is invaluable in applications requiring fast queries within a specific task range, such as real-time adaptive systems and design optimization. Reducing the number of collocation points and training time is a crucial factor in this context. Another promising application of PINN adaptivity is function discovery, where a pre-trained PINN, initially trained on an easy-to-solve PDE, is fine-tuned using sparse measurements obtained from experiments or sensors, as demonstrated by Chen et al. (2021).

### Adaptivity in Inverse Problems

The idea of adaptive PINNs extends to inverse problems, where they excel at inferring unknown parameters and minimizing data dependencies. In these cases, efficient model adaptivity is key in reducing the number of data samples needed to infer initial or boundary conditions, making PINNs especially valuable for real-time adaptive systems. For example, Xu et al. (2022) demonstrated the effective use of transfer learning in PINNs for real-world problems like tunneling, where the model is pre-trained on an easily solvable task and then fine-tuned using limited real-world data, highlighting their effectiveness in inverse applications.

### Comparison with Conventional Solvers

Despite their adaptive capabilities, PINNs must be rigorously evaluated as forward solvers by direct comparisons with conventional numerical methods. Studies such as Qin et al. (2022) have shown that while PINNs offer a flexible, mesh-free, and data-free approach that is beneficial for complex geometries, they often struggle to compete with highly optimized PDE solvers in terms of computational efficiency. For instance, despite leveraging meta-learning to accelerate PINN optimization, Qin et al. (2022) found that their meta-solver remained slower than a strong JAX baseline using the finite-volume method with Godunov flux. This highlights the significant speed advantage that conventional PDE solvers, particularly those implemented in high-performance computing frameworks, can achieve. However, it is important to note that the comparison of Qin et al. (2022) focused on a single benchmark problem, and broader testing across diverse scales and PDE types is needed to fully characterize the strengths and limitations of adaptive PINNs versus optimized traditional solvers. Furthermore, other works surveyed here lack systematic comparisons, limiting conclusive insights into the practical viability of PINNs.

### Emerging Alternatives: Differentiable Solvers and Active Learning

Beyond adaptive PINNs, emerging fast differentiable solvers, combined with active learning, present an alternative for creating surrogate models tailored to specific solution distributions. A fast differentiable solver could iteratively refine a surrogate model by aligning it with a desired solution range, using active learning to prioritize critical regions of the domain. In this context, transfer learning and meta-learning techniques could enhance these approaches by enabling efficient adaptation to new tasks or domains, reducing the need for extensive retraining. However, these methods are still being developed, and their broad adoption remains uncertain. Advancing these methodologies will require systematic comparisons to determine their trade-offs in accuracy, generalization, and computational cost, particularly in relation to adaptive PINNs. Some examples of differentiable solvers include XLB, a differentiable lattice-Boltzmann library (Ataei & Salehipour, 2024); the work of (Pervez et al., 2024) and (Chen et al., 2024), which combines mechanistic

neural networks with a differentiable ODE solver to enable interpretable solutions; and Diffrax, a JAX-based library for numerical differential equation solvers (Kidger, 2021), to name a few. In the context of neural surrogate models, there is recent work exploring active learning Musekamp et al. (2025), which could also be extended to PINNs.

### Benchmarking Challenges and Standardization Needs

Although significant progress has been made, challenges persist that may hinder the continued advancement of adaptive PINNs. One such challenge is the variety of benchmarking strategies used in different studies, which complicates the process of directly comparing methods. Without standardized evaluation frameworks, it becomes difficult to draw meaningful conclusions about the effectiveness of different approaches. In future research, it is necessary to ground these strategies to facilitate easier comparison between different methods. Establishing standardized evaluation guidelines, as proposed in Section 4, is crucial to ensure fair comparisons and evaluating generalization capabilities. The diversity in benchmarking strategies makes it currently difficult to determine the most effective technique. This challenge is reflected in the contradictory conclusions drawn by different works. For example, Penwarden et al. (2023) found that MAML weight initialization only marginally improves performance compared to random initialization, while Qin et al. (2022) and Liu et al. (2022) reported opposite findings. Establishing a common ground for evaluating these techniques is essential. Additionally, refining the definitions of terms such as 'in-distribution', 'out-of-distribution' tasks, and 'related' or 'similar' PDEs can facilitate meaningful comparisons and help identify suitable applications for each method.

### Strategies for Enhancing Adaptivity

Nevertheless, a comparison of adaptivity benefits, using a standard PINN as the baseline, reveals significant improvements in accuracy and training efficiency when adopting weight initialization methods. The full details of this comparison, including the specific equations evaluated, are provided in Appendix B. This comparison underscores the effectiveness of weight initialization approaches in enhancing adaptivity, with the works of Liu et al. (2022), Penwarden et al. (2023), and Cho et al. (2024b) demonstrating particularly promising results in this regard. These methods enable rapid adaptation by leveraging pre-trained models, making them especially well-suited for tasks that require fast adaptation. However, for these methods to be truly effective, their pre-training phase must capture a broad range of shared representations, ensuring generalizability across different tasks.

Similarly, techniques that use basis function expansions, such as those in Chen & Koohy (2024), Cho et al. (2024b), and Gao et al. (2022), offer efficient fine-tuning by reducing the required number of parameters. By restricting the hypothesis space to a smaller set of expressive basis functions, these methods can achieve faster convergence compared to unstructured parameter optimization, provided that the selected bases align well with the task structure. However, like weight initialization methods, shared bases—often frozen during fine-tuning—must be expressive enough to capture a wide variety of tasks.

An alternative strategy to further enhance adaptivity involves incorporating adaptive basis functions or a library of pre-trained basis weights, as suggested by Mustajab et al. (2024). Combining this approach with a mixture-of-experts framework could significantly boost the model's generalization ability. Furthermore, while explicit basis function methods, such as GPT-PINN (Chen & Koohy, 2024) and One-shot PINN Desai et al. (2021), construct global basis functions, exploring smaller local subdomains within the PDE domain may offer further improvements in both efficiency and task-specific adaptability. These strategies highlight potential avenues for future research to improve adaptivity and efficiency in PINNs.

In addition to these approaches, another promising method is the self-referential learning technique introduced by Huang et al. (2022). This approach achieved training runs nearly nine times faster than traditional PINNs when applied to the medium-difficulty Burgers equation and about five times faster for the more complicated Maxwell equations. Their results suggest that using task-specific inputs and keeping the weights trainable hold particular promise for OOD settings, where adaptivity to new tasks is crucial. Moreover, as previously discussed, the adaptive loss function approach of Song et al. (2024) is particularly notable for being one of the few methods benchmarked on the complex Lid-driven cavity flow equation. This method

moves away from traditional weight initialization techniques by dynamically adjusting error weights through an attentional mechanism, demonstrating its potential for challenging fluid dynamics problems. Exploring such innovative loss function strategies could be another promising direction for future research, to further improve optimization and adaptability in PINNs.

**Future Research Directions**

While recent advancements in adaptive PINNs are encouraging, there remain substantial opportunities for further improvement. Future research should focus on developing optimizers tailored for PINNs and more efficient methods for approximating the derivatives of the PDE-loss components, such as those proposed by Shi et al. (2024), to accelerate training. Another promising direction is learning multiple tasks together through parametric PINNs, as explored by Cho et al. (2024a), which could enable a shared representation across tasks and improve generalization. Additionally, creating pre-trained weight libraries tailored to specific problem classes may enhance adaptability and reduce computational overhead. Addressing inefficiencies during adaptation, through strategies to mitigate negative transfer learning, could be crucial to robustness. Incorporating advanced techniques, such as gradient-based attention mechanisms or gradient-weighted loss functions, might dynamically prioritize critical regions within the domain, enabling PINNs to allocate resources more effectively. These advancements are poised to further enhance the adaptability and efficiency of PINNs, making them applicable to a broader range of scientific and engineering problems.

## 6 Conclusion

This study highlights the potential of enhancing the adaptivity of PINNs through meta-learning and transfer learning techniques. By enabling the reuse of learned information, these approaches can improve the efficiency of PINNs, particularly in applications where repeated evaluations of similar tasks are required. Instead of solving each PDE from scratch, adaptive PINNs aim to leverage prior knowledge, making them a promising direction for reducing computational effort in solving families of related problems.

The works presented in this survey support the idea of adaptive PINNs, demonstrating progress while also highlighting areas that require further development. Additionally, this survey provides insights into methodologies for assessing efficient model adaptation, including relevant metrics and benchmarks. Standardizing evaluation practices is essential for facilitating meaningful comparisons across studies and advancing the field.

Adaptive PINNs have the potential to expand the applicability of PINNs in scientific and engineering problems, particularly in scenarios where limited data is available and rapid evaluations are necessary. However, challenges such as computational overhead and generalization across diverse problem domains remain open challenges for further investigation. Continued research in optimizing adaptation techniques, loss function design, and multi-task learning has the potential to further advance adaptive PINNs within the broader landscape of computational methods for PDEs.

## Acknowledgements

We thank the anonymous TLMR reviewers whose feedback helped to improve the manuscript. We acknowledge the support of the German Federal Ministry of Education and Research (BMBF) as part of InnoPhase (funding code: 02NUK078), the Deutsche Forschungsgemeinschaft (DFG, German Research Foundation) under Germany's Excellence Strategy - EXC 2075 – 390740016, and the Stuttgart Center for Simulation Science (SimTech).

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

# A    Benchmark Equations

## A.1    Equation Classification

Table 4 synthesizes the broader landscape of benchmark usage, categorizing partial differential equations (PDEs) by three critical dimensions: computational complexity, key technical challenges, and adoption frequency in the surveyed literature. This structure enables systematic comparison of problem difficulty and helps researchers select appropriate benchmarks for specific evaluation needs.

The **"Complexity"** column rates equations from 1 to 5 stars (☆ to ★★★★★), reflecting relative computational demands. For example, the Poisson equation (1 star) serves as a low-complexity baseline due to its linear, steady-state nature, while the Navier-Stokes equations (5 stars) represent the upper complexity extreme, requiring resolution of coupled velocity-pressure fields with nonlinear advection and complex boundary conditions.

**Key Drivers of Complexity** identify the primary technical challenges for each PDE:

- *Time dependence*: Critical in transient problems (e.g., Burgers', Wave equations)

- *Nonlinearity*: Dominates shock-forming systems (e.g., Allen-Cahn, A-D-R)

- *High-frequency oscillations*: Challenges resolution limits (e.g., Schrödinger, Helmholtz)

- *Multi-variable coupling*: Increases system dimensionality (e.g., Navier-Stokes)

The **"Used By"** column quantifies benchmark prevalence in this survey, revealing distinct adoption patterns. Burgers' equation emerges as the most popular intermediate benchmark (9 studies), offering balanced complexity through its nonlinear shock dynamics. In contrast, high-complexity systems like A-D-R and Navier-Stokes appear only once each, reflecting their specialized computational requirements.

## A.2    Poisson Equation

The Poisson equation is a second-order elliptic PDE appearing in many fields, such as electrostatics, steady heat transfer, and many others. This equation has the following form:

$$
\begin{aligned}
-\Delta u(\boldsymbol{x}) = f(\boldsymbol{x}), &\quad \boldsymbol{x} \in \Omega, \\
u(\boldsymbol{x}) = 0, &\quad \boldsymbol{x} \in \partial\Omega,
\end{aligned}
\tag{9}
$$

where $\Delta(\cdot)$ is the Laplace operator. A common feature among all works is that the domain is 2D, specifically $\Omega \subseteq [-1, 1] \times [-1, 1]$, except for Bischof & Kraus (2022), which uses an L-shaped domain. The forcing or source term $f(\boldsymbol{x})$ changes among works:

## A.3    Burgers' Equation

Burgers' equation is a time-dependent PDE that models a system consisting of a moving viscous fluid. The 1D form of the equation models the fluid flow through an ideal thin pipe. The Burgers' equation is given by:

$$
\begin{aligned}
\frac{\partial u}{\partial t} + u\frac{\partial u}{\partial x} - \nu\frac{\partial^2 u}{\partial x^2} = 0, &\quad x \in \Omega, \quad t \in [0, T], \\
u(x, t) = 0, &\quad x \in \partial\Omega, \quad t \in [0, T], \\
u(x, 0) = u_0(x), &\quad x \in \Omega,
\end{aligned}
\tag{10}
$$

The unknown $u(x, t)$ is the speed of the fluid, and $\nu$ the fluid viscosity. When the viscosity is low, then the fluid flow develops a shock wave.

Most of the works presented here treat $\nu$ as the parameter that defines a task, the initial condition as $u_0(x) = -\sin(\pi x)$, and the computational domain as $\Omega \in [-1, 1]; t \in [0, 1]$. The table below shows the different choices of $\nu$ across various works.

Table 4: Complexity of Different Models

| Model Complexity | | | | |
|---|---|---|---|---|
| **Model** | **Used by** | **Complexity** | **Key Drivers of Complexity** | **Characteristics** |
| Poisson | 5 | ★☆☆☆☆ | Geometric/boundary complexities | Linear, steady-state, single scalar field with fixed boundary values |
| Helmholtz | 2 | ★★☆☆☆ | High-frequency/oscillatory solutions | Linear, oscillatory, high wave numbers require fine-scale resolution |
| Wave | 1 | ★★☆☆☆ | Time dependence | Describes wave propagation and behavior over time |
| Burgers' | 9 | ★★★☆☆ | Nonlinearity, Time dependence, Shocks | Nonlinear advection, shock formation (low viscosity), sharp gradients |
| Allen-Cahn | 3 | ★★★☆☆ | Nonlinearity, Time dependence | Nonlinear reaction-diffusion for phase transitions; stiff, time-dependent dynamics |
| Schrödinger | 3 | ★★★★☆ | High-frequency/oscillatory solutions | Complex-valued solutions, time-dependent, possibly nonlinear |
| A-D-R | 1 | ★★★★★ | Nonlinearity, Time dependence, Stiffness | Combines advection, diffusion, and nonlinear reaction; stiff gradients |
| Navier-Stokes (LDC) | 1 | ★★★★★ | Nonlinearity, Time dependence, Multi-variable | Coupled PDEs (velocity and pressure), nonlinear advection, complex boundaries (sharp corners) |

**Note:** Abbreviations: Advection-Diffusion-Reaction = A-D-R, Lid-driven Cavity = LDC.

Table 5: Different forms of $f(x, y)$ used in various studies. Desai et al. (2021) employs a different forcing term during testing. In the work of Liu et al. (2022), $n$ represents the number of heat sources, and $\mathcal{U}$ denotes uniform sampling.

| **Literature** | $f(x, y)$ | **Parameters** |
|---|---|---|
| Desai et al. (2021) | $\sin(k\pi x)\sin(k\pi y)$ | $k \in \{1, 2, 3, 4\}$ |
| Liu et al. (2022) | $\sum_{i=1}^{n} c_i \cdot \exp\left(-\frac{(x-a_i)^2 + (y+b_i)^2}{0.01}\right)$ | $a_i, b_i \sim \mathcal{U}(0.1, 0.9)$, $c_i \sim \mathcal{U}(0.8, 1.2)$ |
| Bischof & Kraus (2022) | 1 | - |
| Song et al. (2024) | $2\pi^2 \sin(\pi x)\sin(\pi y)$ | - |

Table 6: Different choice of $\nu\, for\, Burgers'\, Equation\, 1D$ used in various studies.

| Literature | Parameters |
|---|---|
| Liu et al. (2022) | $\nu \in [0, 0.1/\pi]$ |
| Penwarden et al. (2023) | $\nu \in [0.005, 0.05]$ |
| Chen & Koohy (2024) | $\nu \in [0.005, 1/\pi]$ |
| Toloubidokhti et al. (2023) | $\nu \in [0.001, 0.1]$ |
| Chen & Koohy (2024) | $\nu \in [0.005, 1/\pi]$ |
| Psaros et al. (2022) | $\nu \in [0.001, 0.002]$ & $\nu \in [0.01, 1.0]$ |
| Song et al. (2024) | $\nu = 0.01/\pi$ |

### A.3.1 Allen-Cahn Equation

The Allen-Cahn equation is given by:

$$
\begin{aligned}
\frac{\partial u}{\partial t} - \lambda \Delta u + \epsilon(u^3 - u) &= f(x,t), & x \in \Omega, & \quad t \in [0,T], \\
u(x,t) &= 1, & x \in \partial\Omega, & \quad t \in [0,T], \\
u(x,0) &= x^2 \cos(\pi x), & x \in \Omega, &
\end{aligned}
\tag{11}
$$

where in $\Omega = [-1,1]$ represents the spatial domain, and $T = 1$ denotes the final time. The coefficient $\lambda$ is chosen from the interval $[0.0001, 0.001]$, while the parameter $\epsilon$, which controls the strength of the nonlinear term, is selected from the range $[1,5]$. The forcing term f(x,t) is set to zero for this study, focusing on the intrinsic dynamics of the Allen-Cahn equation. This represents an initial-boundary value problem (IBVP) as per Chen & Koohy (2024). An alternative formulation of the IBVP exists in other works that derive a forcing term based on an exact solution, such as Penwarden et al. (2023) and Xu et al. (2022).

## A.4 Wave Equation

The wave equation for a scalar wave function $u(x,t)$ is given by:

$$
\frac{\partial^2 u}{\partial t^2} = c^2 \nabla^2 u,
\tag{12}
$$

where $c$ is the wave speed and $\nabla^2$ is the Laplacian operator in three dimensions.

## A.5 Helmholtz Equation

The Helmholtz equation for a scalar field $u(\mathbf{r})$ is given by:

$$
\Delta u + k^2 u = 0,
\tag{13}
$$

where $k$ is the wave number related to the wavelength $\lambda$.

### A.5.1 Schrödinger Equation

The time-dependent Schrödinger equation for a single particle in three-dimensional space is given by:

$$
i\hbar \frac{\partial \Psi(\mathbf{r}, t)}{\partial t} = -\frac{\hbar^2}{2m} \nabla^2 \Psi(\mathbf{r}, t) + V(\mathbf{r}) \Psi(\mathbf{r}, t),
\tag{14}
$$

where $\Psi(\mathbf{r}, t)$ is the wave function, $\mathbf{r} = (x, y, z)$ are the spatial coordinates, $t$ is time, $\hbar$ is the reduced Planck's constant, $m$ is the mass of the particle, $\nabla^2$ is the Laplacian operator, and $V(\mathbf{r})$ is the potential energy function.

### A.6 Advection-Reaction-Diffusion

Advection-reaction-diffusion equations, as considered in this section, are known to be stiff problems when the advection term dominates over the diffusion one. In such cases, sharp transition layers appear in the solution, which are difficult to capture by traditional numerical schemes." Baty (2024) The advection-reaction-diffusion equation is given by:

$$\frac{\partial u}{\partial t} + \mathbf{v} \cdot \nabla u = D\nabla^2 u + R(u), \tag{15}$$

where $u = u(\mathbf{r}, t)$ is the dependent variable (scalar field), $t$ is time, $\mathbf{r} = (x, y, z)$ represents spatial coordinates, $\mathbf{v} = (v_x, v_y, v_z)$ is the velocity field (advection term), $D$ is the diffusion coefficient, $\nabla^2$ is the Laplacian operator, and $R(u)$ is the reaction term.

### A.6.1 Lid-driven Cavity Flow

The lid-driven cavity flow equations are:

$$
\begin{aligned}
\frac{\partial u}{\partial x} + \frac{\partial v}{\partial y} &= 0, \\
u\frac{\partial u}{\partial x} + v\frac{\partial u}{\partial y} &= -\frac{1}{\rho}\frac{\partial p}{\partial x} + \nu\nabla^2 u + F_x, \\
u\frac{\partial v}{\partial x} + v\frac{\partial v}{\partial y} &= -\frac{1}{\rho}\frac{\partial p}{\partial y} + \nu\nabla^2 v + F_y,
\end{aligned}
\tag{16}
$$

with boundary conditions:

$$
\begin{aligned}
u(x, 0) &= 0, \quad u(x, 1) = 1 \quad \text{(lid)}, \\
v(0, y) &= v(1, y) = 0 \quad \text{(walls)}, \\
u(x, y) &= v(x, 0) = v(x, 1) = 0 \quad \text{(other boundaries)}.
\end{aligned}
$$

Here, $u(x, y)$ and $v(x, y)$ are the velocity components, $p(x, y)$ is the pressure, $\rho$ is the fluid density, $\nu$ is the kinematic viscosity, and $F_x$, $F_y$ are additional body forces.

## B  Method Comparison

This table provides a comparison of various PINN methods across different problem types, training approaches, and improvements in accuracy and speed. It highlights the accuracy improvement (%) and speedup/slowdown (%) achieved by each method compared to the baseline PINN. The table includes key details such as the author, short name, problem type, and training strategy (pre-train/fine-tune), along with the data type used in each case. It also reports the end accuracy (for both the PINN and method) and the number of epochs and time required for training. The improvement percentages are calculated based on the following formulas:

$$\text{Accuracy Improvement (\%)} = \left(\frac{\text{PINN}_{\text{accuracy}} - \text{Method}_{\text{accuracy}}}{\text{PINN}_{\text{accuracy}}}\right) \times 100$$

$$\text{Speed Up/Slowdown (\%)} = \left(\frac{\text{PINN}_{\text{epochs}}}{\text{Method}_{\text{epochs}}} - 1\right) \times 100$$

Where speed-up is calculated in terms of either the number of epochs or total training time (* denotes training time).

In terms of Table 7, the following can be observed: TL-gPINN (Lin & Chen, 2024) enhances accuracy over gPINN but remains slower than the baseline PINN. Reptile (Liu et al., 2022) outperforms PINN in the Burgers' inverse problem, achieving faster convergence with fewer epochs. SVD-PINN (Gao et al., 2022) demonstrates a negative performance compared to the standard PINN for the Allen-Cahn equation. Curriculum (Mustajab et al., 2024) accelerates convergence by 50%, though final accuracy is comparable to PINN. Interpolation (Penwarden et al., 2023) improves convergence speed through superior weight initialization but does not significantly boost accuracy. GPT-PINN (Chen & Koohy, 2024) converges more quickly but does not surpass PINN in final accuracy, potentially due to the limited expressiveness of its basis functions. MAD-PINN (Huang et al., 2022) offers a 30% accuracy improvement and a 362% speed-up over PINN for the Maxwell equation, a complex problem. LA-PINN (Song et al., 2024) meta-learns the loss function, outperforming PINN on both simple and complex equations, though the performance gains are less pronounced for more difficult problems. Hyper-lr-PINN (Cho et al., 2024b) excels in both accuracy and speed for the Helmholtz equation, although the speed increase may stem from the similarity between the target and source tasks.

**Note:** The values presented in the table are approximated, as some were extracted from graphical data.

| Author | Short Name | Problem Type | Type | Col. Points | Improvement (%) Accuracy | Improvement (%) Speed | Equation | End Accuracy PINN | End Accuracy Method | Epochs \|\| Time PINN | Epochs \|\| Time Method | Reference |
|---|---|---|---|---|---|---|---|---|---|---|---|---|
| Lin & Chen (2024) | TL-gPINN | Inverse | Curriculum | 40000 | 44 | −84 | Schrödinger (Linear) | $3.40 \times 10^{-5}$ | $1.91 \times 10^{-5}$ | 302* | 1862* | Table 2 |
| Lin & Chen (2024) | TL-gPINN | Inverse | Curriculum | 40000 | 28 | −66 | Schrödinger (Nonlinear) | $3.00 \times 10^{-3}$ | $2.16 \times 10^{-3}$ | 789* | 2293* | Table 3 |
| Liu et al. (2022) | Reptile | Inverse | Weight Init | 2000/200 | 74 | 12000 | Burgers | 56.94 | 14.8 | 60000 | 500 | Table 6 \| Fig.17 |
| Gao et al. (2022) | SVD-PINN | Forward | Weight Init | N/S | −844 | −60 | Allen Cahn | $9.00 \times 10^{-4}$ | $8.50 \times 10^{-3}$ | 8000 | 20000 | Fig.3 |
| Mustajab et al. (2024) | Curriculum | Forward | Curriculum | 512 | 0 | 50 | Wave | $5.00 \times 10^{-6}$ | $5.00 \times 10^{-6}$ | 1500 | 1000 | Fig.9-g |
| Liu et al. (2022) | Reptile | Forward | Weight Init | 4000 | 99 | 1900 | Poisson | $5.07 \times 10^{-5}$ | $6.47 \times 10^{-7}$ | 4000 | 200 | Fig. 6 |
| Liu et al. (2022) | Reptile | Forward | Weight Init | 2000 | 99 | 567 | Burgers | $8.80 \times 10^{-3}$ | $8.50 \times 10^{-5}$ | 2000 | 300 | Table 4 \| Fig. 11 |
| Liu et al. (2022) | Reptile | Forward | Weight Init | 2000 | 98 | 191 | Schrödinger | $6.90 \times 10^{-1}$ | $1.34 \times 10^{-2}$ | 30000 | 10300 | Table 5\| Fig.15 |
| Penwarden et al. (2023) | Interpolations | Forward | Weight Init | 10100 | 12 | 462 | Burgers | $5.10 \times 10^{-3}$ | $4.50 \times 10^{-3}$ | 146 | 26 | Table 1 |
| Penwarden et al. (2023) | Interpolations | Forward | Weight Init | 10100 | 16 | 493 | Heat | $5.50 \times 10^{-3}$ | $4.60 \times 10^{-3}$ | 178 | 30 | Table 3 |
| Penwarden et al. (2023) | Interpolations | Forward | Weight Init | 10100 | 21 | 966 | Allen Cahn | $1.45 \times 10^{-2}$ | $1.15 \times 10^{-2}$ | 469 | 44 | Table 5 |
| Chen & Koohy (2024) | GPT-PINN | Forward | Network | 20200 | −5400 | 140 | Burgers | $2.00 \times 10^{-5}$ | $1.10 \times 10^{-3}$ | 12000 | 5000 | Fig.10-top |
| Huang et al. (2022) | MAD-PINN | Forward | Input | N/S | 0 | 757 | Burgers | $1.00 \times 10^{-2}$ | $1.00 \times 10^{-2}$ | 3000 | 350 | page 8 \| Fig.4 (a) |
| Huang et al. (2022) | MAD-PINN | Forward | Input | N/S | 30 | 362 | Maxwells | 0.04 | 0.028 | 60000 | 13000 | page 8 \| Fig.4 (b) |
| Song et al. (2024) | LA-PINN | Forward | Loss | 10300 | 59 | 25 | Burgers | $6.87 \times 10^{-4}$ | $2.83 \times 10^{-4}$ | 10000 | 8000 | Fig.13 |
| Song et al. (2024) | LA-PINN | Forward | Loss | 8200 | 80 | 900 | Poisson | $2.87 \times 10^{-4}$ | $5.83 \times 10^{-5}$ | 10000 | 1000 | Page 10 \| Fig.8 |
| Song et al. (2024) | LA-PINN | Forward | Loss | 20400 | 99 | 999900 | Helmholtz | $3.83 \times 10^{-2}$ | $2.29 \times 10^{-4}$ | 10000 | 1 | Fig.19 |
| Cho et al. (2024b) | Hyper-LR-PINN | Forward | Weight Init | 11400 | 98 | 19900 | Helmholtz | 1.00 | 0.0285 | 2000 | 10 | Table 24 \| Fig.7 |

Table 7: Comparison of PINN methods. The reference columns refer to the location of where the values were extracted from the source works. Abbreviations; N/S = Not Specified. Negative percentages indicate worse performance compared to baseline PINN.

