# OpenReview forum: "Adaptive Physics-informed Neural Networks: A Survey"
_TMLR — Accepted by TMLR_

### Review · Reviewer_g4RT · 2024-12-10

**Summary Of Contributions:**

The paper is a survey on Physics-Informed Neural Networks that make use of transfer learning and meta-learning algorithms for efficient adaptation to reduce the cost of retraining. The survey begins with a short introduction to PINNs and IBVPs and the similarities between PINNs and numerical methods for solving PDEs. The introduction is followed by a brief description of model adaptivity in Machine Learning focusing on transfer learning, and meta-learning techniques, in particular, parameter efficient fine-tuning and MAML. The core of the paper is an exhaustive list of applications of transfer learning and meta-learning techniques to PINNs aimed at reducing computational cost and improving convergence. The paper ends with a section dedicated to metrics and benchmarks for evaluating adaptivity and efficiency of PINNs, and a section on applications that covers forward and inverse problems.

**Audience:**

Yes

**Broader Impact Concerns:**

There are no concerns on the ethical implications of this work.

**Claims And Evidence:**

Yes

**Requested Changes:**

I understand the difficulty of comparing performances of PINNs but such a comparison, perhaps highlighting failure modes of the various methods, would certainly add value to the survey that, as it is, represents a valuable and comprehensive list of techniques that may be useful for the newcomer but might be of little use for the practitioner.

Could you add some references to section 2.4?

**Strengths And Weaknesses:**

Strengths: perhaps the biggest drawback of vanilla physics-informed neural networks, as pointed out in the paper,  is that they are trained on single PDE instances and have to be retrained from scratch when the PDE changes.  For most scientific and engineering applications this problem makes PINNs impractical. Meta-learning and transfer learning are certainly a promising way to overcome this issue and to the best of my knowledge there is no comprehensive survey that puts all the methods together. Furthermore, I find the PINNs literature particularly difficult to navigate due to the many papers that are published on many different journals/conferences spanning different fields. Therefore, surveys and literature reviews can be very valuable.

Weaknesses: Generally, while such a collection of relevant works can be valuable for the reason discussed above, it can be of little practical utility for the practitioner since it does not directly compare performances of the methods described. At least those that are comparable. With PINNs it is generally hard to get an idea of what works best and for what setting. A truly helpful survey facilitates the job of practitioners when faced with the choice of what algorithms to use.

Minor comments: I don't fully understand the purpose of Section 2.3 and the connection to numerical methods that is described. Could you explain better the radar chart? I don't understand what the values are.

---

> ### Author Response · Authors · 2025-02-05
>
> We appreciate the reviewer’s insightful feedback regarding the challenges of performance comparisons in PINN methodologies. The corresponding weaknesses and requested changes are detailed below.
>
> >**W1:** Generally, while such a collection of relevant works can be valuable for the reason discussed above, it can be of little practical utility for the practitioner since it does not directly compare performances of the methods described. At least those that are comparable. With PINNs it is generally hard to get an idea of what works best and for what setting. A truly helpful survey facilitates the job of practitioners when faced with the choice of what algorithms to use.
> >**RC1:** I understand the difficulty of comparing performances of PINNs but such a comparison, perhaps highlighting failure modes of the various methods, would certainly add value to the survey that, as it is, represents a valuable and comprehensive list of techniques that may be useful for the newcomer but might be of little use for the practitioner.
>
> We acknowledge this concern. Due to variations in benchmark problems and evaluation strategies, direct comparisons are challenging. However, a table has been added in the appendix listing works that directly benchmarked against the vanilla PINN baseline using common PDEs (as detailed in [appendix B]). Additionally, recommendations on promising approaches are discussed in the [Applications & Discussion] section. It should be noted that this table provides only a rough estimate, as some values were extracted from graphs while others were sourced from tables and text.
>
> >**W2:** Minor comments: I don't fully understand the purpose of Section 2.3 and the connection to numerical methods that is described. Could you explain better the radar chart? I don't understand what the values are.
>
> R1: The purpose of Section 2.3 is to establish connections between PINNs and existing numerical methods. For example, the collocation method minimizes residuals, aligning with the optimization principle of PINNs. Similarly, surrogate modeling extracts basis functions from a set of solutions and reuses only the most relevant ones, reducing dimensionality—a concept akin to transfer learning.
>
> A more detailed explanation of the radar chart has been provided.

---

> > ### Comment · Reviewer_g4RT · 2025-02-12
> >
> > Thank you for addressing my concerns.

---

> > > ### Author Response · Authors · 2025-02-19
> > >
> > > Thank you for your feedback. We appreciate your time and effort in reviewing our work. Please let us know if there is anything further we can address.

---

### Review · Reviewer_otXa · 2024-12-30

**Summary Of Contributions:**

The manuscript surveys adaptability in physics-informed neural networks, focusing on transfer learning and meta-learning. After providing an introductory overview of the relevant concepts, the survey collects, summarizes, and groups the relevant PINN literature in several transfer and meta-learning categories. Additionally, sections 4 and 5 briefly discuss metrics, benchmark problems, and applications used and addressed in the adaptive PINN literature.

The main contribution of this work is the survey itself -- there is no new evaluation or significant interpretation of the existing body of work. The contributions are assessed in more detail below.

**Audience:**

Yes

**Broader Impact Concerns:**

This work does not require an ethics statement.

**Claims And Evidence:**

Yes

**Requested Changes:**

**Errors and suggestions**:
- Throughout the manuscript you repeatedly refer to "data generation". I couldn't understand what exactly this means in the different contexts. It seems it is sometimes used as a synonym for a "solver". I would strongly recommend against such a use or, if I misunderstand, to explain more precisely what you mean by "data generation".
- "data resources" -> data requirements (page 2)
- (optional) "To the best of our knowledge, there has not been a review paper addressing model adaptivity in PINNs." This is a reviewer-pleasing statement which I would exclude from the final version.
- Eq. 3: the residuals are missing the squares (or some norm in general). Furthermore, why are the residuals in bold? These are just scalars (there is some inconsistency with eq 2). Lastly, the statement that "weights act as a regularization term" is wrong.
- Also, the statement "optimizing ... leads the solution $u_\theta$ to converge toward the true solution" is not true -- there is no such convergence guarantee. I get what you mean, but I'd recommend being more precise here.
- (optional) Eq 6: consider defining what $\omega(x)$ is (presumably a positive scalar function over the same domain).
- "either continuous or piecewise" (page 6). Piecewise basis functions can and almost always are continuous. Perhaps you mean globally vs locally supported functions.
- (optional) You introduce POD as Proper Orthogonal Decomposition only after you have used the abbreviation POD several times.
- Eq 7: you must start the summation at $i=1$ to retain $N$ modes (otherwise it's $N+1$).
-  (optional) Sec. 2.6: Please define computational efficiency similar to how you did with data efficiency. Does this refer to pre-training, fine-tuning, or also inference?
- (important) PEFT: in the broader ML community PEFT was developed and refers to fine-tuning of truly large models. I would not recommend referring to any kind of partial weight fine-tuning as PEFT. I would sincerely recommend renaming this section/method or at the very least clarifying what you understand under PEFT.
- Page 10: "leveraging on knowledge" -> "leveraging the knowledge"
- Sec 3.2 first paragraph has multiple errors.
- Page 11, last sentence "A" -> "The"
- Page 12: "reptile" -> "Reptile".
- Page 12: "a SVD" -> "an SVD"
- Page 12: "future work will address" ->  "future work should address". Similar issues in other paragraphs (e.g. "future research is concerned with")
- There are many more small errors, so please run the entire manuscript through a writing tool.


**Problematic claims**:
- "This characteristic poses a barrier to the widespread adoption of PINNs across scientific and engineering applications." While adaptability certainly is a barrier, there are other major problems with PINNs (which are also discussed in the introduction). This phrasing can be misinterpreted as adaptability being the primary issue. (similarly with "this survey seeks to identify promising directions for future research to enable the widespread adoption of PINNs")
- "these methods can help overcome the convergence challenges typically associated with PINNs" Convergence on complex problems is one of the central issues of PINNs, which has motivated lots of research on this. I believe this is a very bold claim, and while the surveyed papers indicate some potential, I would expect a more critical assessment of this claim, either in isolation or in relation to other approaches that deal with convergence.
- "synthesizing insights to establish a foundation for efficient data generation strategies tailored to PINNs" Similar to the major criticism, I do not think this survey succeeds at synthesizing new insights, certainly not foundational ones. A slight paraphrasing could fix this.
- "While both neural operators and PINNs have strengths and limitations, this study suggests that PINNs are more suitable for data generation tasks in scientific and engineering domains, especially in applications with limited data resources." Nothing in the main body of text discusses NOs vs PINNs, certainly not to an extent where informed suggestions could be made. I'd suggest to paraphrase this.
- "To achieve efficient model adaptation, this work suggests the application of transfer learning and meta-learning to physics-informed neural networks." Similar to the above points. It *surveys* the application of ...


The primary quality criterion of TMLR is how well-supported the claims of the submission are. I sincerely invite the authors to assess this feedback and consider reformulating the phrasing where necessary to more accurately reflect their intent and the scope of the work.

**Strengths And Weaknesses:**

**Strengths**
- Overall, the topic and the work are well-structured and easy to follow.
- Most concepts are introduced to a sufficient level of detail.
- The survey seems comprehensive/exhaustive (although I would like the authors to clarify this in the rebuttal and the manuscript).
- Sections 2.3 - 2.5 on the adaptability in classical reduced-order modeling are an original perspective in the PINN literature.
- Altogether, this serves as a valuable entry point to adaptive PINNs.

**Weaknesses**
- My primary criticism of this work is that it merely collects, groups, and summarizes existing work. Although this is one function of a survey paper, I think the authors overlook an opportunity to offer more to the reader with a stronger discussion. For example, while model adaptivity is motivated repeatedly throughout the work, there is no critical discussion or limitations of the individual works or adaptive PINNs overall. There are no comparisons (qualitative or quantitative) or recommendations on which methods to choose and when. The future works are mostly repeated from the individual papers missing the opportunity to summarize where the entire field is or should be going. In my view, the discussion in Section 6 is currently much too short and superficial, especially given that the paper is 17 pages long.
- Connecting this criticism to the contributions listed by the authors on page 3, I support 1 (overview) and 2 (survey) as notable contributions, but 3 (metrics and benchmarks) and 4 (future directions) are not significant enough to be distinguished from 2.
- There are quite a few presentation issues in terms of style and grammar, but also mathematical errors and missing or confusing explanations. However, I expect most of these to be fixed (see the list below).
- This might be due to phrasing, but some claims are borderline unjustified (see below).

---

> ### Author Response · Authors · 2025-02-05
>
> Thank you to the reviewer for the thoughtful and constructive feedback. We recognize the importance of providing a deeper discussion, critical analysis, and meaningful comparisons to guide future research.
>
> >**W1:** My primary criticism of this work is that it merely collects, groups, and summarizes existing work. Although this is one function of a survey paper, I think the authors overlook an opportunity to offer more to the reader with a stronger discussion. For example, while model adaptivity is motivated repeatedly throughout the work, there is no critical discussion or limitations of the individual works or adaptive PINNs overall. There are no comparisons (qualitative or quantitative) or recommendations on which methods to choose and when. The future works are mostly repeated from the individual papers missing the opportunity to summarize where the entire field is or should be going. In my view, the discussion in Section 6 is currently much too short and superficial, especially given that the paper is 17 pages long.
>
> We agree that more profound synthesis and actionable insights are essential for a survey paper. To address this, we strengthen the discussion by incorporating a more refined critique and deeper insights into the surveyed works:
>
> - [Appendix A.1] now includes a structured comparison of benchmark problems, categorizing them by complexity (e.g., PDE type, dimensionality, smoothness).
> - [Appendix B] provides a comparative analysis of methods that benchmark against standard PINNs, and common benchmark problems are provided. Note that this is quite difficult since several works are benchmarked differently.
> - [Section 5 - Discussion] We expand on the limitations of existing approaches, offer a comparative evaluation of adaptive methods, and provide recommendations on when specific techniques are most effective. Future research directions are also refined beyond the perspectives presented in individual papers, offering a more comprehensive outlook on where the field is heading.
>
> - [Appendix A.1] now includes a structured comparison of benchmark problems, categorizing them by complexity (e.g., PDE type, dimensionality, smoothness).
> - [Appendix B] provides a comparative analysis of methods that benchmark against standard PINNs, and common benchmark problems are provided. Note that this is quite difficult since several works are benchmarked differently.
> - [Section 5 - Discussion] We expand on the limitations of existing approaches, offer a comparative evaluation of adaptive methods, and provide recommendations on when specific techniques are most effective. Future research directions are also refined beyond the perspectives presented in individual papers, offering a more comprehensive outlook on where the field is heading.
>
> >**W2:** Connecting this criticism to the contributions listed by the authors on page 3, we agree with the significance of Contributions 1 (overview) and 2 (survey), but Contributions 3 (metrics and benchmarks) and 4 (future directions) are not sufficiently distinctive from Contribution 2.
>
> - **Metrics and Benchmarks**: We have added a comparison of various benchmark equations and provided recommendations for error measurements in evaluating multi-task generalization.
>
> >**W3:** There are several presentation issues, including style, grammar, and mathematical errors, along with missing or unclear explanations. However, we expect most of these issues to be resolved (please see the list below).
>
> To improve readability and correctness, we have carefully revised the manuscript to correct grammatical errors and enhance the clarity and accuracy of the mathematical formulations.
>
> >**W4:** This may be a phrasing issue, but some claims are borderline unjustified (see below).
>
> We have re-assessed the claims for accuracy, refined overgeneralized statements, and included additional supporting references where necessary.
>
> >**RC1:** Throughout the manuscript, you repeatedly refer to "data generation." I am unclear on what exactly this means in different contexts. It seems it is sometimes used as a synonym for a "solver." I would strongly recommend against this usage or, if I misunderstand, please explain more precisely what you mean by "data generation."
>
> To avoid confusion, we have removed the term "data generation" and replaced it with "solution approximation" for clarity. [Abstract & Line 42]
>
> >**RC2:** "data resources" -> "data requirements" (page 2).
>
> This change has been made.
>
> continued...

---

> > ### Author Response · Authors · 2025-02-05
> >
> > >**RC3:** (optional) "To the best of our knowledge, there has not been a review paper addressing model adaptivity in PINNs." This is a reviewer-pleasing statement, which I would recommend excluding from the final version.
> >
> > This statement has been removed.
> >
> > >**RC4:** Eq. 3: The residuals are missing the squares (or some norm in general). Furthermore, why are the residuals in bold? These are just scalars (there is some inconsistency with eq. 2). Lastly, the statement that "weights act as a regularization term" is incorrect.
> >
> > - The residuals are now squared, and the notation has been corrected. [Equation 3]
> > - "Regularization" has been replaced with "loss balancing term" for better accuracy. [Lines 126–128]
> >
> > >**RC5:** The statement "optimizing ... leads the solution uθ to converge toward the true solution" is not true — there is no convergence guarantee. I understand the intent, but I recommend being more precise here.
> >
> > We have reworded it: "The network aims to learn a solution that approximates the true solution." [Lines 129–131]
> >
> > >**RC6:** (optional) Eq. 6: Consider defining what ω(x) is (presumably a positive scalar function over the same domain).
> >
> > We have added a definition for ω(x) to address this. [Lines 142-144]
> >
> > >**RC7:** "either continuous or piecewise" (page 6). Piecewise basis functions can, and almost always are, continuous. Perhaps you mean globally vs. locally supported functions.
> >
> > This has been removed, and instead, we now highlight the relevant properties of basis functions. [Lines 149–152]
> >
> > >**RC8:** (optional) You introduce POD as Proper Orthogonal Decomposition only after you have used the abbreviation POD several times.
> >
> > This has been fixed. [Lines 161–162]
> >
> > >**RC9:** Eq. 7: You must start the summation at i=1 to retain N modes (otherwise it's N+1).
> >
> > This issue has been corrected. [Equation 7]
> >
> > >**RC10:** (optional) Sec. 2.6: Please define computational efficiency in a manner similar to how you defined data efficiency. Does this refer to pre-training, fine-tuning, or also inference?
> >
> > We have added a definition for computational efficiency. [Lines 211-214]
> >
> > >**RC11:** (important) PEFT: In the broader ML community, PEFT was developed and refers to the fine-tuning of truly large models. I would not recommend referring to any kind of partial weight fine-tuning as PEFT. I recommend renaming this section/method or at the very least clarifying what you understand by PEFT.
> >
> > We have added a specification to clarify what PEFT means in this context, which involves aspects of model adaptivity, a key idea of this work. [Lines 232-236 & 283-290]
> >
> > >**RC12 -17:** Typos / Grammar
> >
> > We have addressed all typos and grammar issues using writing software.
> >
> > >**PC1:** "This characteristic poses a barrier to the widespread adoption of PINNs across scientific and engineering applications." While adaptability certainly is a barrier, there are other major problems with PINNs (which are also discussed in the introduction). This phrasing can be misinterpreted as adaptability being the primary issue. (Similarly with "this survey seeks to identify promising directions for future research to enable the widespread adoption of PINNs.")
> >
> > We have rephrased this in the abstract to: "However, limitations related to convergence and the need for re-optimization with each change in PDE parameters hinder their widespread adoption across scientific and engineering applications." [abstract]
> >
> > >**PC2:** "These methods can help overcome the convergence challenges typically associated with PINNs." Convergence on complex problems is one of the central issues of PINNs, which has motivated much research in this area. I believe this is a very bold claim, and while the surveyed papers indicate some potential, I would expect a more critical assessment of this claim, either in isolation or in relation to other approaches that deal with convergence.
> >
> > We have reworded this to: "In addition, these methods show potential in addressing some of the convergence challenges associated with PINNs." [Line 83]
> >
> > >**PC3:** "Synthesizing insights to establish a foundation for efficient data generation strategies tailored to PINNs." Similar to the major criticism, I do not think this survey succeeds at synthesizing new insights, certainly not foundational ones. A slight paraphrasing could fix this.
> >
> > We have rephrased this to: "This survey highlights the idea of efficient model adaptivity for PINNs and its potential to facilitate broader adoption in real-world applications where data are scarce and fast evaluation is essential." This better reflects the purpose of the paper. [Lines 83-84]
> >
> > continued...

---

> > > ### Author Response · Authors · 2025-02-05
> > >
> > > >**PC4:** "While both neural operators and PINNs have strengths and limitations, this study suggests that PINNs are more suitable for data generation tasks in scientific and engineering domains, especially in applications with limited data resources." Nothing in the main body of the text discusses NOs vs PINNs to an extent where informed suggestions could be made. I'd suggest paraphrasing this.
> > >
> > > We have rephrased this to: "Considering the challenges associated with data acquisition in scientific and engineering domains, this study suggests the use of PINNs..." This avoids directly comparing NOs vs. PINNs. [Lines 53-61]
> > >
> > > >**PC5:** "To achieve efficient model adaptation, this work suggests the application of transfer learning and meta-learning to physics-informed neural networks." Similar to the above points. It surveys the application of ...
> > >
> > > We have rephrased this to: "To achieve efficient model adaptation, this work surveys the application of transfer learning and meta-learning to physics-informed neural networks." [Lines 219-220]
> > >
> > > We believe these revisions address all concerns and significantly enhance the paper’s rigor and value to the community.

---

> ### Comment · Reviewer_otXa · 2025-02-13
>
> I thank the authors for considering and successfully incorporating the feedback. I particularly appreciate the updated sections 4 and 5 making this a more complete survey. I think the structure and content of section 4 is much improved, especially with the support of the appendix. Section 5 now has a more comprehensive and critical discussion and I appreciate the connections to broader trends in PINNs and scientific ML.
>
> I would merely suggest adding more format structure to section 5, such as paragraph titles, since this section is now a rather long wall of text. Although the grammar is much improved, I still notice a few typos, so I suggest another thorough read.
>
> With this, I will update my recommendation.

---

> > ### Author Response · Authors · 2025-02-19
> >
> > Thank you for your positive feedback and for recognizing the improvements in Sections 4 and 5. We appreciate your suggestion regarding Section 5's structure and will add paragraph titles to improve readability. We will also conduct another thorough proofreading to address any remaining typos. Please let us know if there is anything else we can clarify or improve.

---

### Review · Reviewer_m2K5 · 2025-01-01

**Summary Of Contributions:**

This paper is a review of adaptive physics informed neural network models. Physics informed networks are now well-known models for solving forward and inverse problems involving partial differential equations. However, they are limited by the need for reoptimization with changes in PDE parameters.
This indicates a need for models that can adapt to changes in parameters and this paper reviews a number of such models.

The paper is well organized. It begins with an introduction of PDE methods dividing them into traditional solvers, neural PDE solvers and neural surrogates (such as neural operators), with the advantage of neural PDE solving (such as PINNs) being that they include physical laws in learning. However PINNs are not naturally adaptive.

Section 2 of the paper provides an overview of physics-informed networks and related them to classical collocation methods. Section 2.4 also discusses a classical adaptation method, reduced order modeling and discusses their relation with PINNs. Section 2.7 and 2.8 provide a brief introduction to transfer learning and meta-learning.

Section 3 comprises the core of the paper where various transfer learning and meta learning methods for PINNs are discussed. The transfer learning methods are divided into methods with full fine-tuning and methods with parameter-efficient fine tuning with a brief section on curriculum learning. Meta-learning methods follow the taxonomy of Hospedales et al (2021), dividing the models into 1) models that learn the weight initialization, 2) models that learn the network structure, 3) models that learn the loss function, 4) models that learn the input and 4) few-shot learning models.

Section 4 presents benchmarks and metrics; section 5 has a discussion of applications and the paper ends with a conclusion in section 6.

In my opinion the paper presents a useful overview of the state-of-art for adaptive physics-informed networks. There are however some concerns of clarity in discussion of some referenced works (see below), a full discussion of benchmarks and applications is missing.

**Audience:**

Yes

**Broader Impact Concerns:**

No broader impact concerns.

**Claims And Evidence:**

Yes

**Requested Changes:**

The concerns mentioned above regarding clarity and missing discussion in referenced works should be addressed.

Furthermore the sections on benchmarks, metrics and application need significant changes. I think that a critical discussion of applications is important for this paper.

**Strengths And Weaknesses:**

In the remaining review I mainly focus on sections 3, 4 and 5. Since this is a review paper, for evaluation I judge whether for each reviewed work the reader is able to get  a high-level view of what is done, why and how it was done and what was the conclusion. Opaque descriptions that provide no insight should be avoided.

Meta-learning history box:

Since a number of methods later described depend on MAML there should be a description of how MAML works. Currently the description only says that it “aimed to provide a good weight initialization”. Later there is an indirect reference to MAML’s bi-level optimization and second order derivatives in the context of the Reptile algorithm. This should be restructured.

Section 3.1.1.

The TL-PINN description is good. It might be useful to also include the source tasks.

 The 3rd paragraph mentions TL-gPINNS as an extension of gPINN, but I can’t find any earlier mention of gPINN  nor is there any citation. The description should introduce gPINNs first and then the extension.

The description of S-FEM method is not clear about the motivation of the method. What is the significance of the FEM data pre-training? And how is it different from other types of pre-training?

Section 3.1.2

The description of the method from Desai et. al. (2021) is unclear in how it relates to the PINN objective. What is the relation of the ‘final solution’ as it is given in equation 8 to the PINN objective from section 2.2 and equations 2 and 3? Is \Phi_\theta the same as u_\theta? I think the description should use the PINN notation from section 2.2.

The third paragraph (page 10) retraining for each displacement step is mentioned but is unclear what a displacement step is.

The description of the methods from Goa et al (2022) and Pellegrin et. al. (2022) are good.

In the last paragraph discussing Xu et al. (2022) what are the offline and online stages? Are these the same as pre-training and fine-tuning or something else? Is there streaming data involved?

Section 3.2.1

I think this section should first include a general description of how the meta-learning initialization techniques are applied to PINNs. Without such a general description the individual methods are very opaque and it is not clear how the methods work.

The third paragraph mentions that the implementations of Zhong et al and Cheng & Alkhalifah differ without describing how they are different.

The fourth paragraph mentions LEAP without a description of LEAP.

The fifth paragraph mentions Penwarden et al. comparing several models. What were the conclusions of this comparison and the two-step initialization?

In the last paragraph what were the conclusions from Cho et al?

3.2.2, 3.2.3, 3.2.4 have good descriptions of the relevant methods. However, one paragraph at the beginning summarizing the approaches would be helpful such as the one at the beginning of section 3.2.4.

Section 3.3

I think the sentence “Few-shot learning aims to minimize the number of training examples required…” should be reconsidered, since few-shot learning is more than just sample efficient learning.

The last two paragraphs are not very informative about the works discussed. For instance, what is the method of Mustajab (2024) and how does it achieve few-shot learning? Similarly for the other methods there is not sufficient discussion.

Sections 4.1, 4.2

There is no discussion of the character (difficulty,complexity, scale etc) and comparison of the various benchmark problems. Similarly there is no discussion on the usefulness or otherwise of the various error metrics that are used in the literature.

Section 5.

The discussion on application is also very sparse and only quotes various applications from the papers. I think it is missing a more critical evaluation of the types of applications for which adaptive models are useful and those which they are not.
A full illustrative example of the benefits of adaptivity would be very useful and would make the discussion concrete.

Typos/Grammar:

Transfer learning box: “Sharkley”

Transfer learning box: “Pan” reference missing.

Transfer learning box: “Zhuang et al.” citation format.

Meta learning box: “work of Schmihuber (1987), work”

2.6: ‘Burgers’ apostrophe

3.2: ‘study of this techniques’

3.2.1: ‘adoptation’

3.2.1: page 12 ‘...Xavier in a unsupervised…’

---

> ### Author Response · Authors · 2025-02-05
>
> We sincerely thank the reviewer for taking the time to provide detailed and insightful feedback.
>
> >**W1:** Since a number of methods later described depend on MAML there should be a description of how MAML works. Currently the description only says that it “aimed to provide a good weight initialization”. Later there is an indirect reference to MAML’s bi-level optimization and second order derivatives in the context of the Reptile algorithm. This should be restructured.
>
>
> We have now provided a more detailed description of MAML and Reptile in the meta-learning history box. See lines \[ Meta-learning history box \]
>
> >**W2:** The TL-PINN description is good. It might be useful to also include the source tasks.
>
> We provide a more specific description of the equation characteristics in the footnote and a description of the source task in [Lines 263-266].
>
> >**W3:** The 3rd paragraph mentions TL-gPINNS as an extension of gPINN, but I can’t find any earlier mention of gPINN nor is there any citation. The description should introduce gPINNs first and then the extension.
>
> We restructured the paragraph first to introduce gPINNs, followed by the proposed method. [Lines 362-374]
>
> >**W4:** The description of S-FEM method is not clear about the motivation of the method. What is the significance of the FEM data pre-training? And how is it different from other types of pre-training?
>
> We emphasize the role of S-FEM in the work. Generally, pre-training on S-FEM data leads to higher accuracy compared to pre-training on FEM data due to the superior quality of data produced by S-FEM. [Lines 274-277]
>
> >**W5:** The description of the method from Desai et al. (2021) is unclear in how it relates to the PINN objective. What is the relation of the ‘final solution’ as it is given in equation 8 to the PINN objective from section 2.2 and equations 2 and 3? Is $\Phi_\theta$ the same as $\mathbf{u_\theta}$ I think the description should use the PINN notation from section 2.2.
>
> To avoid confusion, we standardize the notation according to section 2.2. [Eq 8 and Lines 291-306]
>
> >**W6:** The third paragraph (page 10) retraining for each displacement step is mentioned but is unclear what a displacement step is.
>
>
> We add clarification regarding the displacements. [Lines 311-313 & Footnote]
>
> >**W7:** In the last paragraph discussing Xu et al. (2022) what are the offline and online stages? Are these the same as pre-training and fine-tuning or something else? Is there streaming data involved?
>
> We standardize the terminology to "pre-training" and "fine-tuning" instead of “offline” and “online” stages [Lines 378 & 380]
>
> >**W8:**  I think this section should first include a general description of how the meta-learning initialization techniques are applied to PINNs. Without such a general description the individual methods are very opaque and it is not clear how the methods work.
>
> We add the suggested description to section 3.2.1. [Lines 390-397]
>
> >**W9:** The third paragraph mentions that the implementations of Zhong et al and Cheng & Alkhalifah differ without describing how they are different.
>
> We add a distinction between both methods. [Lines 407-411]
>
> >**W10:** The fourth paragraph mentions LEAP without a description of LEAP.
>
> We include an explanation and reference to LEAP. [Lines 424-428]
>
> continued...

---

> > ### Author Response · Authors · 2025-02-05
> >
> > > **W11:** The fifth paragraph mentions Penwarden et al. comparing several models. What were the conclusions of this comparison and the two-step initialization?
> >
> > We clarify the method and add the conclusions of the work. [Lines 435-445]
> >
> > > **W12:** In the last paragraph what were the conclusions from Cho et al?
> >
> > We include the conclusions from the work of Cho et al. [Lines 453-458]
> >
> > >**W13:** 3.2.2, 3.2.3, 3.2.4 have good descriptions of the relevant methods. However, one paragraph at the beginning summarizing the approaches would be helpful such as the one at the beginning of section 3.2.4.
> >
> > We add introductory paragraphs at the beginning of sections 3.2.2 and 3.2.3. [Lines 460-464 & 498-504]
> >
> > >**W14:** I think the sentence “Few-shot learning aims to minimize the number of training examples required…” should be reconsidered, since few-shot learning is more than just sample efficient learning
> >
> > We revise the paragraph to include a broader definition of few-shot learning. [Lines 567-577]
> >
> > >**W15:** The last two paragraphs are not very informative about the works discussed. For instance, what is the method of Mustajab (2024) and how does it achieve few-shot learning? Similarly for the other methods there is not sufficient discussion.
> >
> > We emphasize how the works mentioned in this section use few-shot learning. [Lines 589-617]
> >
> > > **W16:** There is no discussion of the character (difficulty,complexity, scale etc) and comparison of the various benchmark problems. Similarly there is no discussion on the usefulness or otherwise of the various error metrics that are used in the literature.
> >
> > We add a table in the appendix categorizing the difficulty and detailing the characteristics of the standard benchmark problems used in the survey. [Lines 624-636 & Appendix A.1]. We also detail the importance of error metrics, the need for strategies for generalization, and the necessity of including a similarity measure for tasks when reporting errors in multi-task settings. [Lines 638-660]
> >
> > >**W17:** The discussion on application is also very sparse and only quotes various applications from the papers. I think it is missing a more critical evaluation of the types of applications for which adaptive models are useful and those which they are not. A full illustrative example of the benefits of adaptivity would be very useful and would make the discussion concrete.
> >
> > We merge the Applications section with the discussion to avoid redundancy and address applications more broadly, incorporating insights from related works benchmarking real-life applications. We also add more depth to the discussion and future work perspectives. [Section 5]
> >
> > >**W18:** Typos/Grammar.
> >
> > Thank you. We have addressed all typos and grammar issues.
> >
> > >**RC1:** Furthermore the sections on benchmarks, metrics and application need significant changes. I think that a critical discussion of applications is important for this paper.
> >
> > We address this by improving the discussions on applications, benchmarks, and error metrics, as outlined above.

---

> > > ### Comment · Reviewer_m2K5 · 2025-02-12
> > >
> > > Thank you for the incorporating the suggested changes.
> > >
> > > However, I still believe that a critical experimental evaluation and comparison of the various methods is quite important for users of the methods surveyed in this paper, as has also been pointed out by the other reviewers otXa and g4RT.
> > >
> > > What would improve the paper is an experimental comparison of chosen methods for various applications where adaptive PINNs would be useful and a critical review of advantages and shortcomings.
> > > This aspect is still missing from the current version of the paper. The new table on page 30 in the appendix is a start but this needs be more integrated with then main paper.

---

> > > > ### Author Response · Authors · 2025-02-19
> > > >
> > > > Thank you for your thoughtful feedback and for recognizing the improvements to the manuscript. We appreciate your suggestion regarding an experimental comparison of the surveyed methods.
> > > >
> > > > We believe that running experiments to compare all methods on several datasets is beyond the scope of a survey paper.  However, we have aimed to provide a comprehensive overview by summarizing existing studies and highlighting general trends in adaptive PINN methods.
> > > >
> > > > The table summarizing applications has been discussed in the main text to offer insights into the conclusions that can be drawn from the approaches that were directly comparable. It remains in the appendix due to the lack of available data for definitive quantitative comparisons.
> > > >
> > > > We have addressed all other comments and remain open to further refinements as needed.

---

> > > > > ### Comment · Reviewer_m2K5 · 2025-02-19
> > > > >
> > > > > Thank you for the response. I think what you suggest is fair given that the paper is a survey of methods. My concerns and suggestions regarding clarity also were resolved. I will update my recommendation.

---

### Decision · Action_Editor_tmiH · 2025-02-25

**Recommendation:** Accept with minor revision

**Comment:**

A cataloguing of various techniques is described, including fine-tuning, meta-learning, transfer learning, learning the loss function, the inputs and so on. Metrics and benchmarks are discussed in detail. The text is very clearly written, and the total amout of references is in the order of 70. All reviewers agree that this is a good submission, and while initially they were sceptical, they also upgraded their scores.

Reading the paper, I noticed that some recent published works on 'Mechanistic Neural Networks' are not mentioned. I think they are quite relevant and mentioning them in the surve would make sense.

https://dl.acm.org/doi/10.5555/3692070.3693713
https://openreview.net/forum?id=Oazgf8A24z

All in all, I think this is an important survey that is currently missing from the literature.

**Audience:**

Yes, very much so. Physics-Informed Neural Networks are very hot, and in my humble opinion, they will soon become quite relevant for perception also (beyond sciences and engineering).

**Claims And Evidence:**

This is a survey paper on adaptive Physics-Informed Neural Networks. The focus is on scientific and engineering applications. There are no claims and evidence per se, although the text go overs about 70 relevant references from the recent state-of-the-art.